# Emergent Symbol-like Number Variables in Artificial Neural Networks

**Satchel Grant**                                             *grantsrb@stanford.edu*
*Departments of Psychology and Computer Science*
*Stanford University*

**Noah D. Goodman**                                          *ngoodman@stanford.edu*
*Departments of Psychology and Computer Science*
*Stanford University*

**James L. McClelland**                                        *jlmcc@stanford.edu*
*Departments of Psychology and Computer Science*
*Stanford University*

**Reviewed on OpenReview:** *https://openreview.net/forum?id=YPnYpiru5W*

## Abstract

What types of numeric representations emerge in neural systems, and what would a satisfying answer to this question look like? In this work, we interpret Neural Network (NN) solutions to sequence based number tasks through a variety of methods to understand how well we can interpret them through the lens of interpretable Symbolic Algorithms (SAs)—precise algorithms describable by rules operating on typed, mutable variables. We use GRUs, LSTMs, and Transformers trained using Next Token Prediction (NTP) on tasks where the correct tokens depend on numeric information only latent in the task structure. We show through multiple causal and theoretical methods that we can interpret raw NN activity through the lens of simplified SAs when we frame the neural activity in terms of neural subspaces rather than individual neurons. Using Distributed Alignment Search (DAS), we find that, depending on network architecture, dimensionality, and task specifications, alignments with SA's can be very high, while other times they can be only approximate, or fail altogether. We extend our analytic toolkit to address the failure cases by expanding the DAS framework to a broader class of alignment functions that more flexibly capture NN activity in terms of interpretable variables from SAs, and we provide theoretic and empirical explorations of Linear Alignment Functions (LAFs) in contrast to the preexisting Orthogonal Alignment Functions (OAFs). Through analyses of specific cases we confirm the usefulness of causal interventions on neural subspaces for NN interpretability, and we show that recurrent models can develop graded, symbol-like number variables within their neural activity. We further show that shallow Transformers learn very different solutions than recurrent networks, and we prove that such models must use anti-Markovian solutions—solutions that do not rely on cumulative, Markovian hidden states—in the absence of sufficient attention layers.

## 1 Introduction

We can see examples of the modeling power of Neural Networks (NNs) in both biological NNs (BNNs) from the impressive capabilities of human cognition, and in artificial NNs (ANNs) where recent advances have had such great success that ANNs have been crowned the "gold standard" in many machine learning communities (Alzubaidi et al., 2021). The inner workings of NNs, however, are still often opaque. This is, in part, due to their representations being highly distributed. Individual neurons can play multiple roles within a network in what's called population encoding. In these cases, human-interpretable information is encoded across

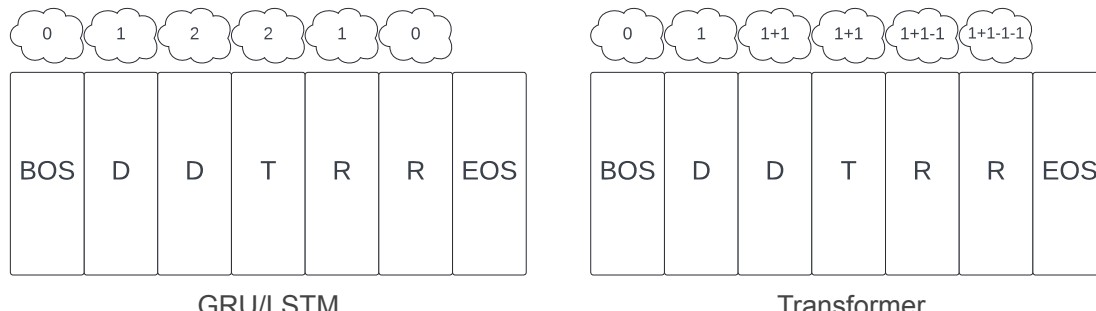

Figure 1: Different architecture's solutions achieving the same accuracy on a numeric equivalence task. The rectangles represent tokens for a task in which the model must produce the same number of R tokens ending with the EOS token as it observed D tokens. The T token indicates the end of the D tokens (see Methods 3.1). The thought bubbles represent the values of causally discovered neural variables encoded within the models' representations. The recurrent models encode a single count variable that increments up before the T token and down after the T token, with 0 indicating the end of the task. Transformers learn a solution in which they recompute the task relevant information from the input tokens at each step in the sequence. All NoPE transformers align with the displayed Transformer solution. RoPE transformers can partially rely on positional information unless they are trained on a variant of the task that breaks number-position correlations.

populations of neurons rather than within any individual unit (Hinton et al., 1986; Smolensky, 1988; Olah et al., 2017; 2020; Elhage et al., 2022; Scherlis et al., 2023; Olah, 2023).

Symbolic Algorithms (SAs), in contrast, defined as processes that manipulate distinct, typed entities according to explicit rules and relations, can have the benefit of consistency, transparency, and generalization when compared to their neural counterparts. A concrete example of an SA is a computer program, where the variables are distinct, typed, mutable entities, able to represent many different values, processed by well defined functions. There are many existing theories that posit the necessity of algorithmic, symbolic, processing for higher level cognition (Do & Hasselmo, 2021; Fodor & Pylyshyn, 1988; Fodor, 1975; 1987; Newell, 1980; 1982; Pylyshyn, 1980; Marcus, 2018; Lake et al., 2017). Human designed symbolic cognitive systems, however, can lack the expressivity and performance of NNs. This is apparent in the field of natural language processing where neural architectures trained on vast amounts of data (Vaswani et al., 2017; Brown et al., 2020; Kaplan et al., 2020) have swept the field, surpassing the pre-existing symbolic approaches. Despite the differences between NNs and SAs, it might be argued that NNs actually implement simplified SAs; or, they may approximate them well enough that seeking neural analogies to these simplified SAs would be a powerful step toward an interpretable, unified understanding of complex neural behavior. In one sense, this pursuit is trivial for ANNs, in that ANNs are, by definition, aligned to the computer program that defines them. The algorithms they acquire through training, however, are not so trivial to understand, and their complexity can be so great that simplified SAs become useful for explaining, predicting, and controlling their behavior. This approach of seeking to characterize NNs (either biological or artificial) in terms of simplified, interpretable SAs is one way of construing the goal of many research programs in cognitive science, neuroscience, and mechanistic interpretability.

In this work, we narrow our focus to numeric cognition and ask, how can we understand neural implementations of numeric values at the level of SAs? Numeric reasoning has the advantage of being well studied in humans of different ages and experience levels, which provides a powerful domain for comparisons between BNNs and ANNs (Di Nuovo & Jay, 2019). And numeric domains provide the benefit of tasks built upon well defined variables. We focus on a numeric equivalence task that was used to test the numeric abilities of humans whose language lacks explicit number words (Gordon, 2004). The task is formulated as a sequence of tokens, requiring the subject to produce the same number of response tokens as a quantity of demonstration tokens initially observed at the beginning of the task. This task is interesting for computational settings because the training labels vary in both identity and sequence length, and quantities are never explicitly labeled. Similar

versions of this task have also been used in previous theoretical and computational work (El-Naggar et al., 2023; Weiss et al., 2018; Behrens et al., 2024), providing a platform to expand our understanding of seemingly disparate modeling systems in unified ways.

What sorts of representations do ANNs use to solve such a task and how do they arrive at these representations? Do the networks represent numbers in a shared system, or do they use different systems for different situations? Is it propitious to think about their representations as though they are discrete variables in an SA, or would it be better to think of their neural activity on a graded continuum? Do the answers to these questions change over the course of training, and do the answers vary based on task and architectural details? How can we unify the way we understand NN solutions in satisfying ways for cognitive scientists, neuroscientists, and computer scientists alike? We set out to understand NN neural activity through the lens of simplified, interpretable SAs using causal interventions to support our interpretations.

In this work, we pursue these questions by training Gated Recurrent Units (GRUs)(Cho et al., 2014), Long Short-Term Memory cells (LSTMs) (Hochreiter & Schmidhuber, 1997), and Transformers on numeric equivalence tasks using Next Token Prediction (NTP). We then provide causal, correlational, and theoretical analyses such as activation patching, Principal Component Analysis (PCA), attention visualizations, and Distributed Alignment Search (DAS) (Geiger et al., 2021; 2023) to understand the networks' representations and solutions, and we introduce the notion of an Alignment Function to the DAS framework which we use to frame neural activity in terms of functions of interpretable variables. We summarize our contributions as follows:

1. We show through causal interventions the emergence of graded symbol-like neural variables in RNNs, where neural variables are defined as representational subspaces that causally align with a variable from an SA, and they are "graded" or "symbol-like" when they exhibit signatures of a continuum rather than being fully discrete.

2. We show that seemingly insignificant task variations can drastically affect the NN's alignment to the SAs, motivating us to extend the DAS framework to a broader class of alignment functions, including linear alignment functions (LAFs). We show that LAFs can be used to control NN behavior through their neural activity aligned to interpretable SAs, and we provide a theoretical analysis of LAFs and Orthogonal Alignment Functions (OAFs) to better understand their properties.

3. We show empirically that shallow Transformers use an anti-Markovian solution to the numeric tasks, and we show theoretically that Transformers must use anti-Markovian solutions in all tasks in the absence of sufficient attention layers.

4. Through our specific analyses, we confirm the utility of interpreting NNs at the level of neural subspaces and using causal interventions to make claims about NN solutions.

We use our results to encourage the use of multiple causal interpretability tools for any representational analysis, to highlight functional differences that might emerge from architectural/task constraints, and to demonstrate the nuances of interpreting neural activity at the level of SAs.

## 2 Related Work

Many prior works have attempted to describe ANNs through SAs. Some relevant examples include Lindner et al. (2023) who built a system to compile human written code into transformer models and Michaud et al. (2024) who perform linear and symbolic regression on simplified NN representations to generate programs that perform like the NNs. Both of these cases focus on generating sufficient SAs for network behavior rather than relating the internal mechanisms of an existing NN to the resulting SAs. Another prominent example is the work of Nanda et al. (2023) who showed that small transformers trained on modular addition use discrete Fourier transforms combined with trig identities to solve the task. Nanda et al. used a combination of examinations and ablations to defend their claims.

We wish to highlight the importance of using causal manipulations for interpreting NN solutions rather than relying on correlational analyses. Causal inference broadly refers to processes that isolate causal effects of individual components within a larger system (Pearl, 2010). An abundance of causal interpretability

variants have been used to determine what functions are being performed by NN models' activations and circuits (Olah et al., 2018; 2020; Wang et al., 2022; Geva et al., 2023; Merrill et al., 2023; Bhaskar et al., 2024; Wu et al., 2024). Vig et al. (2020) provides an integrative review of the rationale for and utility of causal mediation in NN analyses. We rely and build upon DAS for many of our analyses. DAS can be thought of as a specific type of activation patching (also referred to as causal tracing) (Meng et al., 2023; Vig et al., 2020; Heimersheim & Nanda, 2024). In section 3.4, we introduce the notion of generalized *alignment functions* which can be thought of as a specific class of mapping models from Ivanova et al. (2022) where the mapping model in our case defines an invertible, causal relationship between neural subspaces from an NN and interpretable variables from an SA; the success of the relationship is measured through NN behavior. Other works such as Williams et al. (2021) have explored differences between orthogonal and linear relationships between distributed representations where they largely focused their analyses on formally defining metrics on representational dissimilarity.

Many publications explore ANNs' abilities to perform counting tasks (Di Nuovo & McClelland, 2019; Fang et al., 2018; Sabathiel et al., 2020; Kondapaneni & Perona, 2020; Nasr et al., 2019; Zhang et al., 2018; Trott et al., 2018) and closely related tasks (Csordás et al., 2024). Our tasks and modeling paradigms differ from many of these publications in that numbers are only latent in the structure of our tasks without explicit teaching of distinct symbols for distinct numeric values. El-Naggar et al. (2023) provided a theoretical treatment of Recurrent Neural Network (RNN) solutions to a parentheses closing task, and Weiss et al. (2018) explored Long Short-Term Memory RNNs (LSTMs) (Hochreiter & Schmidhuber, 1997) and Gated Recurrent Units (GRUs) (Cho et al., 2014) in a similar numeric equivalence task looking at the activations. These works showed correlates of a magnitude scaling solution in both theoretical and practically trained ANNs. Our work builds on their findings by using causal methods for our analyses, expanding the models considered, and introducing new theoretical and empirical analyses. Behrens et al. (2024) explored transformer counting solutions in a task similar to ours. Our work extends beyond theirs by exploring positional encodings, avoiding explicit labels of the numeric concepts, using causal analyses, and providing a theoretical focus on anti-Markovian solutions.

## 3 Methods

### 3.1 Numeric Equivalence Tasks

Each task we consider is defined by a Next-Token Prediction (NTP) task over sequences, as in the example shown in Figure 1. The goal of the task is to reproduce the same number of response tokens as demonstration tokens observed before the Trigger (T) token. Each sequence starts with a Beginning of Sequence (BOS) token and ends with an End of Sequence (EOS) token. Each sequence is generated by first uniformly sampling an object quantity from the inclusive range of 1 to 20 where 20 was chosen to match the human experiments of Pitt et al. (2022). The sequence is then constructed as the combination of two phases. The first phase, called the demonstration phase (**demo phase**), starts with the BOS token and continues with a series of demo tokens equal in quantity to the sampled object quantity. The end of the demo phase is indicated by the trigger token after the demo tokens. This also marks the beginning of the response phase (**resp phase**). The resp phase consists of a series of resp tokens equal in number to the demo tokens. After the resp tokens, the end of the sequence is denoted by the EOS token.

During the teacher forced, NTP model training, we include all tokens in the NTP loss. During model evaluation and DAS trainings, we only consider tokens in the resp phase—which are fully determined by the demo phase. During model trainings, we hold out the object quantities 4, 9, 14, and 17 as a way to examine generalization. We chose 4, 9, 14, and 17 to semi-uniformly cover the space of possible numbers while including even, odd, and prime numbers and ensuring that training covered 3 examples at both ends of the training range. A trial is considered correct when all resp tokens and the EOS token are correctly predicted by the model after the trigger. We include three variants of this task differing only in their demo and resp token instances.

**Multi-Object Task:** there are 3 demo token instances $\{D_1, D_2, D_3\}$ with a single response token instance, R. The demo tokens are uniformly sampled from the 3 possible instances. An example input sequence with

an object quantity of 2 could be: "BOS $D_3$ $D_1$ T", with a ground truth response of "R R EOS". All possible tokens are contained in the set {BOS, $D_1$, $D_2$, $D_3$, T, R, EOS}.

**Single-Object Task:** there is a single demo token instance, D, and a single response token instance, R. An example of the input sequence with an object quantity of 2 is: "BOS D D T", with a ground truth response of "R R EOS". All possible tokens are contained in the set {BOS, D, T, R, EOS}.

**Same-Object Task:** there is a single token instance, C, used for both the demo and resp tokens. An example of the input sequence with an object quantity of 2 is: "BOS C C T", with a ground truth response of "C C EOS". All possible tokens are contained in the set {BOS, C, T, EOS}.

For some transformer trainings, we include **Variable-Length (VL)** variants of each task to break count-position correlations. In these variants, each token in the demo phase has a 0.2 probability of being sampled as a unique "void" token type, V, that should be ignored when determining the object quantity of the sequence. The number of demo tokens will still be equal to the object quantity when the trigger token is presented. As an example, consider the possible sequence with an object quantity of 2: "BOS V D V V D T R R EOS".

## 3.2 Model Architectures

The recurrent models in this paper consist of Gated Recurrent Units (GRUs) (Cho et al., 2014), and Long Short-Term Memory networks (LSTMs) (Hochreiter & Schmidhuber, 1997). These architectures both have a Markovian, hidden state vector that bottlenecks all predictive computations following the structure:

$$h_{t+1} = f(h_t, x_t) \tag{1}$$

$$\hat{x}_{t+1} = g(h_{t+1}) \tag{2}$$

Where $h_t$ is the hidden state vector at step $t$, $x_t$ is the input token at step $t$, $f$ is the recurrent function (either a GRU or LSTM cell), and $g$ is a multi-layer perceptron (MLP) used to make a prediction, denoted $\hat{x}_{t+1}$, of the token at step $t + 1$.

We contrast the recurrent architectures against transformer architectures (Vaswani et al., 2017; Touvron et al., 2023; Su et al., 2023) in that the transformers use a history of input tokens, $X_t = [x_1, x_2, ..., x_t]$, at each time step, $t$, to make a prediction:

$$\hat{x}_{t+1} = f(X_t) \tag{3}$$

Where $f$ now represents the transformer architecture. We show results from 2 layer, single attention head transformers that use No Positional Encodings (NoPE) (Haviv et al., 2022) and Rotary Positional Encodings (RoPE) (Su et al., 2023). Refer to Supplemental Figure 5 for more model and architectural details. We also consider one-layer transformers with No Positional Encodings (NoPE) in Results section 4.2.2. For all of our analyses except the training curves in Figure 3, we first train the models to $> 99\%$ accuracy on their respective tasks before performing analyses. The models are evaluated on 15 sampled sequences of each of the 16 trained and 4 held out object quantities. We train 5 model seeds for each training condition. One seed from the transformer models in both the Variable-Length Multi-Object and Variable-Length Same-Object tasks were dropped without replacement due to their low accuracy.

## 3.3 Symbolic Algorithms (SAs)

In this work, we examine the alignment of 3 different SAs to the models' distributed representations.

**Up-Down Program:** uses a single numeric variable, called the **Count**, to track the difference between the number of demo tokens and resp tokens at each step in the sequence. It also contains a **Phase** variable to determine whether it is in the demo or resp phase. The program ends when the Count is equal to 0 during the resp phase.

**Up-Up Program:** uses two numeric variables—the **Demo Count** and **Resp Count**—to track quantities at each step in the sequence. It uses a Phase variable to track which phase it is in. This program increments the Demo Count during the demo phase and increments the Resp Count during the resp phase. It ends when the Demo Count is equal to the Resp Count during the resp phase.

**Context Distributed (Ctx-Distr) Program:** queries a history of inputs at each step in the sequence, assigns a numeric value to each, and sums their values to determine when to stop (contrasted against encoding a cumulative, Markovian quantity variable). More specifically, this program uses an Input Value variable for each input token, and assigns the Input Value a value of 1 for demo tokens and -1 for resp tokens and computes the sum of the Input Values at each step in the sequence to determine the count. This program outputs the EOS token when the sum is 0 and the sequence contains the T token.

We include Algorithms 1, 2, and 3 in the supplement which show the pseudocode used to implement the Up-Down, Up-Up, and Ctx-Distr programs in simulations. Refer to Figure 1 for an illustration of the Up-Down strategy and the Ctx-Distr strategy that is observed in some transformers.

It is important to note that there are an infinite number of causally equivalent implementations of these SAs. For example, the Up-Down program could immediately add and subtract 1 from the Count at every step of the task in addition to carrying out the rest of the program as previously described. We do not discriminate between programs that are causally indistinct from one another in this work.

### 3.4 Distributed Alignment Search (DAS)

DAS measures the degree of alignment between a representational subspace from an NN and a symbolic variable from a symbolic algorithm (SA) by testing the assumption that the model hidden state $h \in R^{d_m}$ can be written as an orthogonal rotation $z = Qh$, where $Q \in R^{d_m \times d_m}$ is orthonormal, $z \in R^{d_m}$ consists of contiguous subspaces encoding high-level variables from SAs, and $d_m$ is the size of the hidden state. The benefit of this alignment is that it allows us to understand the NN's activity through interpretable variables and it allows us to manipulate the value of these variables without affecting other information.

Concretely, DAS performed on the Up-Down program tests the hypothesis that $z$ is composed of subspaces $\vec{z}_{\text{count}}$ encoding the Count, $\vec{z}_{\text{phase}}$ encoding the Phase, and $\vec{z}_{\text{extra}}$ encoding extraneous, irrelevant activity.

$$z = \begin{bmatrix} \vec{z}_{\text{count}} \\ \vec{z}_{\text{phase}} \\ \vec{z}_{\text{extra}} \end{bmatrix} \tag{4}$$

Each $\vec{z}_{\text{var}} \in R^{d_{\text{var}}}$ is a column vector of potentially different lengths satisfying the relation $d_{\text{count}} + d_{\text{phase}} + d_{\text{extra}} = d_m$. Under this assumption, the value of a high-level variable encoded in $h$ can be freely exchanged through causal interventions using:

$$h^v = Q^{-1}((1 - D_{\text{var}})Qh^{trg} + D_{\text{var}}Qh^{src}) \tag{5}$$

Where $D_{\text{var}} \in R^{d_m \times d_m}$ is a manually chosen, diagonal, binary matrix with $d_{\text{var}}$ non-zero elements used to isolate the dimensions that make up $\vec{z}_{var}$, $h^{src}$ is the *source vector* from which the subspace activity is harvested, $h^{trg}$ is the *target vector* into which activity is substituted, and $h^v$ is the resulting intervention vector that we then use to replace $h^{trg}$ in the model's processing, allowing the model to make predictions based on a replaced value of variable v*ar* following the intervention.

DAS relies on the notion of counterfactual behavior to create intervention data to train and evaluate $Q$. For a given SA, we know what the SA's behavior will be after performing a causal intervention on one of its variables. The resulting behavior from the SA after intervening on a specific variable and keeping everything else in the algorithm and task constant is the counterfactual behavior. This counterfactual behavior can be used as a training signal for $Q$ using next-token prediction. $Q$ can equivalently learn any row permutation of the subspaces in $z$, thus we can restrict our searches to values of $D_{\text{var}}$ that have contiguous non-zero entries. We can then brute-force search over independent trainings with different values of $d_{\text{var}}$, selecting the $(Q, D_{\text{var}})$ pair with the best results. Unless otherwise stated, we try values of $d_{\text{var}}$ equal to either 16 or half of $d_m$ and take the better performing of the two. See Supplemental Figure 8 for a closer examination of how $d_{\text{var}}$ affects results.

We perform our causal interventions on individual time steps in the sequence. We run the model up to an independently sampled timestep $t$ in the target sequence, taking its latent representation at that point as the target vector, $h_t^{trg}$. We do the same for the source vector, $h_u^{src}$, at timestep $u$ from a separate source

sequence. We then construct $h_t^v$ using Equation 5, and continue the model's predictions starting from time $t$, using $h_t^v$ in place of $h_t^{trg}$.

For the LSTM architecture, we perform DAS on a concatenation of the $h$ and $c$ recurrent state vectors (Hochreiter & Schmidhuber, 1997). In the GRUs, we operate on the recurrent hidden state. In the transformers, we operate on the residual stream following the first transformer layer (referred to as the Layer 1 Hidden States in Supplementary Figure 5) or the input embedding layer. We use 10,000 intervention samples for training and 1,000 samples for validation and testing. For all data, we uniformly sample trial object quantities, and unless otherwise stated, we uniformly sample intervention time points, $t$ and $u$, from sequence positions containing demo tokens or response tokens (excluding BOS, trigger, and EOS tokens). We orthogonalize the rotation matrix using PyTorch's orthogonal parameterization with default settings. We train $Q$ with a batch size of 512 until convergence, selecting the checkpoint with the best validation performance for analysis. We use a learning rate of 0.001 and an Adam optimizer. See more detail in Supplement A.4.

**DAS Evaluation:** Once our rotation matrix has converged, we can evaluate the quality of the alignment using the accuracy of the model's predictions on the counterfactual outputs on held out intervention data. We consider a trial correct when all deterministic tokens are predicted correctly using the argmax over logits. We report the proportion of trials correct as the Interchange Intervention Accuracy (IIA) (as used in previous work (Geiger et al., 2023)).

**DAS Alignment Functions:** In an effort to understand the solutions employed by the Same-Object RNNs, we introduce relaxations of the orthogonal rotation matrix used in DAS. We do this by substituting the orthogonal matrix $Q$ with a general invertible function $f(h)$. We name this class of invertible DAS functions *alignment functions* due to their potential to "align" or encode the relationship between the neural activity and the specified interpretable variables. Formally, we can write the model's aligned representation, $z$, in terms of an invertible function, $f$, where $z = f(h)$. In this work, we only extend DAS to linear cases of $f$ of the form $f(h) = X(h + b)$ where $X \in R^{d_m \times d_m}$ is an invertible symmetric matrix and $b \in R^{d_m}$ is a bias vector. Using $\phi \in \{trg, src\}$ to denote that the same alignment function is applied to both the target and source vectors before the intervention, we reformulate Equation 5 in terms of $f$:

$$z^\phi = f(h^\phi) = X(h^\phi + b) \tag{6}$$
$$h_t^v = X^{-1}((1 - D_{\text{var}})z_t^{trg} + D_{\text{var}}z_u^{src}) - b \tag{7}$$

With this formulation, we are able to train $X$ and $b$ using the same counterfactual sequences used to train $Q$ in Equation 5. We refer to the original DAS analyses as using an **Orthogonal Alignment Function** (OAF) and the linear formulation from Equations 6 and 7 as the **Linear Alignment Function** (LAF). In our experiments, we construct $X = (MM^\top + \epsilon I)S$ where $M \in R^{d_m \times d_m}$ is a matrix of learned parameters initially sampled from a centered gaussian distribution with a standard deviation of $\frac{1}{d_m}$, $I \in R^{d_m \times d_m}$ is the identity matrix, $\epsilon = 0.1$ to prevent singular values equal to 0, and $S \in R^{d_m \times d_m}$ is a diagonal matrix to learn a sign for each column of $X$ using diagonal values $s_{i,i} = \text{Tanh}(a_i) + \epsilon(\text{sign}(\text{Tanh}(a_i)))$ where each $a_i$ is a learned parameter and $\epsilon = 0.1$ to prevent 0 values.

### 3.4.1 Activation Substitutions

**RNN Individual Activation Substitutions:** We explore direct substitutions of individual neurons in the Multi-Object trained RNN models to demonstrate the relative ineffectiveness of individual neuron patching compared to the rotated subspace patches used in DAS. In these experiments, we attempt to transfer the Count in accordance with the Up-Down program by directly replacing the activation value of a single neuron (A.K.A. dimension) in the RNN's hidden state vector at time step $t$ with the activation value of the same neuron at time step $u$ from a different sequence. This is equivalent to Equation 5 using an identity rotation with a single, non-zero value in D corresponding to the index of the specified neuron. We perform these interventions individually for every model neuron and evaluate the model's IIA using the expected behavior from the Count interventions.

**Transformer Hidden State Substitutions:** A sufficient experiment to determine whether a Transformer is using Markovian states is to examine its behavior after replacing all activations in its most recent hidden state vector from time $t$ from a target sequence with a hidden state vector from time $u$ from a different

source sequence. If the post-intervention behavior matches that of the source sequence after time $u$, then the state has encoded all behaviorally relevant information in its activation vector and we can conclude that the intervened transformer state is Markovian. If the post-intervention behavior ignores the substitution and matches the target sequence after time $t$, then we can conclude that the states are anti-Markovian. In the two-layer transformers used in this work, we only need to perform this intervention on the output (A.K.A residual stream or hidden states) of the first transformer layer, as the output of the second layer can no longer transmit information between token positions (see Supplement A.5 for more details). See Supplement A.4.6 for specific intervention data examples.

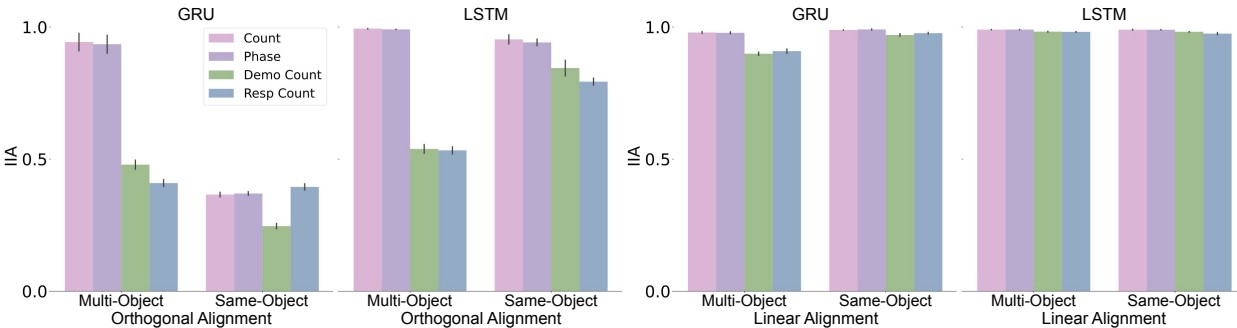

Figure 2: The Interchange intervention accuracy (IIA) for variables from different SAs for different tasks and architectures. The displayed IIA for the Count and Phase variables comes from the Up-Down program. The IIAs for the Demo Count and Resp Count variables come from the Up-Up program. IIA measurements show the proportion of trials where the model correctly predicts all counterfactual R and EOS tokens following a causal intervention. The DAS alignment function is displayed below each panel.

# 4 Results

## 4.1 Recurrent Neural Networks

### 4.1.1 Numeric Neural Variables

The left side of Figure 2 shows the DAS alignments using the orthogonal alignment function for RNNs trained on the Multi-Object and Same-Object tasks. In the Multi-Object recurrent models, we see that the most aligned SA is the Up-Down program from the higher IIA in the Count and Phase variables compared to the Demo Count and Resp Count variables from the Up-Up program. We use this as supporting evidence for the claim that the Multi-Object GRUs and LSTMs possess neural variables for the Count and Phase of the task and use a solution that is causally consistent with the Up-Down program. A visualization of the first two principal components of the solution found by a Multi-Object GRU in Supplemental Figure 16 further supports this interpretation. During the demonstration phase the projected states take steps in one direction along a line; at the trigger token, the trajectory jumps in a direction perpendicular to this line; and during the response phase the states travel parallel to the demonstration phase trajectory, albeit in the reverse direction.

The existence of these numeric neural variables stands as a proof of principle that neural systems do not require explicit exposure to discrete numeric symbols, nor do they need built-in counting principles to develop symbol-like representations of numbers. Such variables can emerge in these architectures simply from learning to perform an exact quantity matching task.

We now focus on the Same-Object task where the demo tokens and resp tokens share a single token id within the task. We see from Figure 2 that each of the Same-Object GRU OAF alignments have relatively low IIA. To aid in understanding how the model solves this task variant, we show the first two principal components of the hidden states of a Same-Object GRU in Supplemental Figure 17. As before, the states progress along a line in one direction during the demonstration phase, and they jump to a line paralleling the first when transitioning to the response phase. In this case, however, the jump returns back to the start of the parallel

line and then proceeds along this line in the same direction as in the demonstration phase; furthermore, the rate of progress and the stopping point in the response phase both depend on the object count of the trial. Thus, the main PCs of this model cannot be simply described in terms of either the Up-Down or Up-Up SAs. We include an additional DAS analysis in Supplement A.3 in which we introduce a new SA that more closely describes our observations from the PCA in Figure 17. These additional alignments, however, also have a relatively low IIA of 58.1% using an OAF.

We considered whether the state spaces of the Same-Object models could be aligned to an SA using an LAF instead of the OAF. We see in Figure 2 that the LAF IIA for the Same-Object GRUs reaches very high accuracy (>95%) for the Count and Phase variables of the Up-Down program. In accordance with the OAF IIA and what we observed in Figure 17, we do not conclude that the model has distinct Count and Phase variables in the same way as the Multi-Object models. It does mean, however, that such variables are recoverable from its state space via an invertible linear mapping, and can be causally intervened upon after such recovery, before being mapped back into the state space of the model to produce desired, predictable behavior.

We also see in Figure 2 that the GRU LAF alignments reach very high IIA (>95%) for the Demo Count and Resp Count variables of the Up-Up program. It appears that the flexibility of the LAFs allows the states of networks to be aligned with conceptually distinct programs. In this case, it is possible to compose the Count variable of the Up-Down program using the difference between the Demo Count and Resp Count variables from the Up-Up program. This perhaps explains why both LAF alignments result in high IIA for both the Multi- and Same-Object tasks as shown on the right side of Figure 2. We show in Supplemental Section A.3 that LAFs do not achieve high IIA for *all* possible SAs, where the LAF IIA only reaches 72.2% and 89.4% for the Multi-Object and Same-Object GRUs respectively.

In sum, LAFs can allow for causal interventions of high accuracy under a wider range of circumstances than OAFs, though, this measure alone is perhaps less diagnostic of the specific SA that best aligns with the NN's learned solution. We return to LAFs in Sections 4.1.5 and 4.1.6 to better explore their theoretic and empirical properties.

### 4.1.2 Individual Activation Substitutions

We performed direct substitutions of individual activation values in recurrent models' hidden state vectors to demonstrate the importance of operating on a subspace of the neural population rather than on individual neurons. We turn our attention to the raw activation traces in the topmost panel (b) of Figure 4, and note that neurons 12 and 18 (shown in blue and black) have a high correlation with the Count of the sequence. These traces came from an LSTM with $d_m = 20$. In this model, we attempted interchange interventions that transferred the raw activity from both neurons 12 and 18 in an attempt to transfer the value of the Count. These interventions achieved an IIA of 0.399 on the behavior generated from the Up-Down program. Furthermore, we observed no consistent pattern of behavior (i.e. off by one errors) following the interventions. We include this result as a cautionary demonstration that interpreting correlations without interventions can be problematic; and the interventions appear to suggest that interpreting raw NN activations can be misleading.

### 4.1.3 Graded Neural Variables

By increasing the granularity of our analyses, we uncovered a graded effect of the content of the values involved in the interchange interventions. We can see this in Figure 3 (c) and (d). We see a gradience in the IIA, where the interventions have a relatively smooth decrease in IIA when the quantities involved in the intervention are large and when the intervention quantities have a greater absolute difference. This indicates that the neural variables have some level of graded continuity despite the task using fully discrete numeric values. We refer to such neural variables as *graded* or *symbol-like*. This notion of graded, symbol-like variables can be contrasted against cases where the errors are uniformly distributed, independently of the values used in the interventions. It is notable that human behavior in many numeric tasks exhibits similar graded effects, called the size and the distance effect (Moyer & Landauer, 1967). Some researchers note that differential experience may account for these effects (and showed this in neural network simulations) (Verguts et al.,

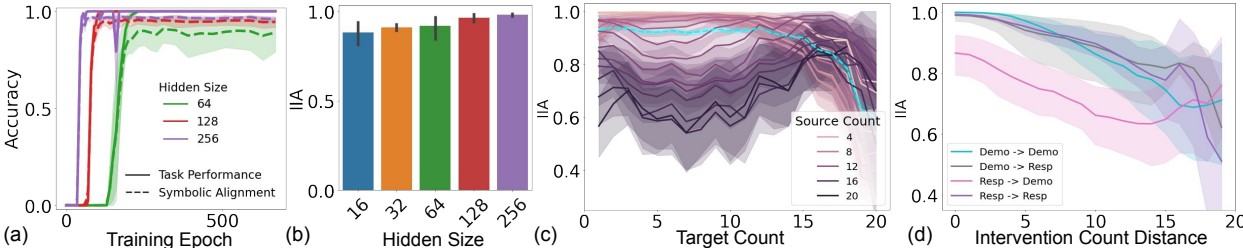

Figure 3: In all panels, the IIA comes from DAS using an Orthogonal Alignment trained on the Count variable in the Up-Down program. The models are all Multi-Object GRUs. (a) Shows task accuracy and IIA over the course of training for architectures with different sizes of the recurrent state $h$. We note the correlation between IIA and accuracy, with relatively little change as training continues. (b) Shows the final IIA for the GRUs as a function of increasing hidden state sizes. (c) Shows the IIA from the 128d GRU as a function of the Count value from the source $h_u^{src}$ (denoted by color) and the Count before the intervention in the target $h_t^{trg}$ (shown on the x-axis). The cyan, dashed line represents the mean IIA over all interventions for a given target count—highlighting the unequal distribution over target source pairs. (d) DAS IIA from the 128d GRU as a function of the absolute difference between the target and source counts. The colors indicate the Phase of $h^{src}$ on the left and $h^{trg}$ on the right. Panels (c) and (d) show that the value of the variable during the interventions somewhat smoothly affect the resulting IIA. See Supplemental Section A.4.7 for detail on the data used in panels (c) and (d).

2005). In our experiments, the task training data provides more experience with smaller numbers, as the models necessarily interact with smaller quantities every time they interact with larger quantities. This is perhaps a causal factor for the greater intervention error at larger numbers, but we do not explore this further. The DAS training data suffers from a similar issue due to the fact that we use a uniform sampling procedure for the object quantities that define the training sequences and we uniformly sample the intervention indices from appropriate tokens in these sequences. This results in a disproportionately large number of training interventions containing smaller values.

The graded neural variables raise the question of how best to interpret neural networks. We remind ourselves that the ANN is built on a symbolic computer program, and thus, this program will always align perfectly with the ANN by definition. The non-trivial goal of our work is to find SAs that characterize the computations such programs learn such that we can predict and control the NNs' activity in unified, interpretable ways. The symbolic gradience that we observe in our models serves as partial motivation for the LAFs examined in Section 4.1.5.

### 4.1.4   Model Width and Developmental Trajectories

**Model Width:** We see in Figure 3(a) that although many model widths can solve the Multi-Object task, increasing the number of dimensions in the hidden states of the GRUs seems to improve the IIA of the Up-Down alignment. We can also see from Figure 3 (b) that the larger models tend to have better IIA. Although our results are for RNNs on linear tasks, an interesting related phenomenon in the LLM literature is the effect of model scale on performance (Brown et al., 2020; Kaplan et al., 2020). We do not concretely explore why increasing dimensionality improves IIA, but we speculate that with greater dimensionality comes a greater likelihood that any two variable subspaces will be orthogonal to one another.

**Developmental Trajectories:** Turning our attention to the learning trajectories in Figure 3, we can see that the models' task accuracy and IIA begin to transition away from 0% at similar epochs and plateau at similar epochs. This finding can be contrasted with an alternative result in which the alignment curves significantly lag behind the task performance of the models. Alternatively, there could have been a stronger upward slope of the IIA following the initial performance jump and plateau. In these hypothetical cases, a possible interpretation could have been that the network first develops more complex solutions, or it could have developed unique solutions for many different input-output pairs and subsequently unified them with further training. The pattern we observe instead is consistent with the idea that the networks are biased

towards simple, unified strategies early in training. Perhaps our result is expected from works like Saxe et al. (2019) and Saxe et al. (2022) which show an inherent tendency for NNs trained via gradient descent to find solutions that share network pathways. This would provide a driving force towards the demo and resp phases sharing the same representation of a Count variable.

### 4.1.5 Understanding Alignment Functions

In this section, we ask why Linear Alignment Functions (LAFs) improve IIA? To answer this, we reformulate the model's neural activity, $h$, in terms of activity component vectors $u_i \in R^{d_m}$: $h = X^{-1}z - b = Uz - b = \sum_{i=1}^{d_m} z_i u_i - b$, where $X$ and $b$ make up the LAF, $U = X^{-1}$ for notational ease, $z$ is a vector composed of interpretable subspaces from Equation 4, and $z_i$ refers to the value of the $i^{\text{th}}$ dimension of $z$. The interchange intervention in Equation 7 is equivalent to exchanging weighted activity components $z_i u_i$:

$$h^v = U(D_{\text{var}}z^{src} + (1 - D_{\text{var}})z^{trg}) - b = -b + \sum_{i=1}^{d_m} \mathbf{1}_{\{i \le d_{var}\}} z_i^{src} u_i + \sum_{i=1}^{d_m} \mathbf{1}_{\{i > d_{var}\}} z_i^{trg} u_i \tag{8}$$

$$h^v = -b + \sum_{i=1}^{d_{\text{var}}} z_i^{src} u_i + \sum_{i=1+d_{\text{var}}}^{d_m} z_i^{trg} u_i = -b + \sum_{i=1}^{d_{\text{var}}} z_i^{src} u_{\text{var},i} + \sum_{i=1+d_{\text{var}}}^{d_m} z_i^{trg} u_{\overline{\text{var}},i} \tag{9}$$

Where $u_{\text{var},i}$ indicates that the activity corresponds to the intervened variable subspace and $u_{\overline{\text{var}},i}$ is all other activity. If $U$ is orthogonal, then by definition, each inner product $\langle u_i, u_j \rangle = 0$ when $i \ne j$. Thus $\langle z_i u_i, z_j u_j \rangle = z_i z_j \langle u_i, u_j \rangle = 0$ too. If $U$ is orthogonal, then so is its inverse, and thus $\langle U^{-1} z_i u_i, U^{-1} z_j u_j \rangle = 0$ when $i \ne j$ due to orthogonal matrices preserving inner products. Thus when using an orthogonal alignment function, the intervened subspaces are also orthogonal in the original neural space. This is not necessarily the case, however, for LAFs, where $U$ is a linear invertible matrix because $\langle U^{-1} z_i u_i, U^{-1} z_j u_j \rangle$ need not be equal

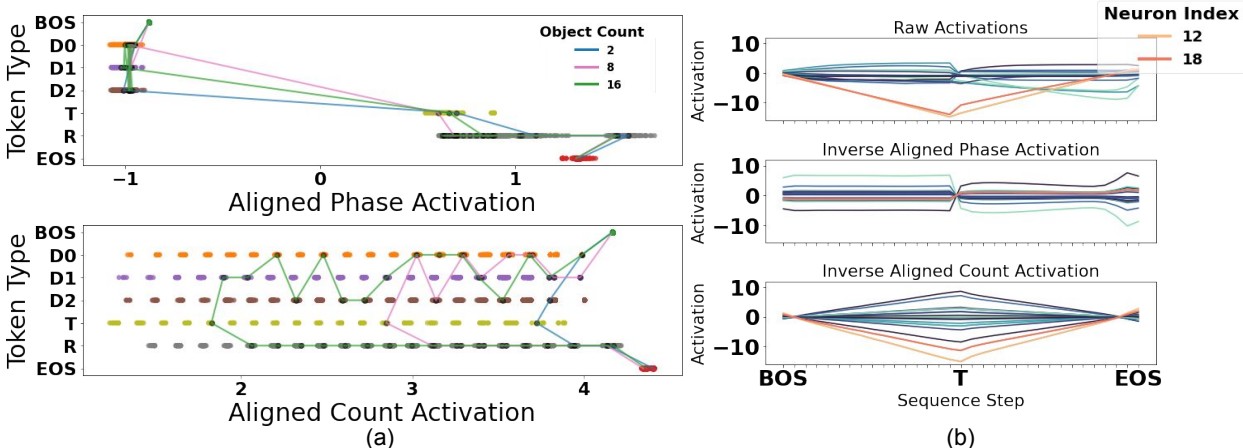

Figure 4: (a) The top panel shows values of $h$ for different trials and input token types projected onto the aligned dimension of a 1D linear alignment, equal to $DX(h+b)$, for the Phase variable. The bottom panel shows the same for the Count variable. The $h$ vectors are collected from 15 trials for each object count ranging from 1-20 from a single Multi-Object LSTM of size 20 (size chosen for relative simplicity of raw activity). The IIA for the Phase was 84.7% and the Count was 82.6%. The connecting lines trace the states from individual trials with object counts of 2, 8, and 16. Dot colors redundantly encode token type. (b) These panels show neural variables in the models' original neural space and show the importance of causal analysis. Each panel shows the activation for individual neurons averaged over 15 trials each with an object count 15. The x-axis shows steps in the trials. The topmost panel shows the raw values. We label two specific neurons (index 12 and 18 within the $h \in R^{20}$ vector) that have a visibly high correlation with the Count of the sequence. We show in Section 4.1.2 that these two neurons are insufficient to causally transfer the Count between representations. The middle and bottom panels show the Phase and Count neural variables projected back into the original neural space by inverting the aligned activity using $X^{-1}(DX(h+b)) - b$.

to 0. This means that the LAF can allow the intervened subspaces to be non-orthogonal in the original neural space, thus allowing the LAF to formulate each variable from the SA in linear terms of the other variables.

We demonstrate this by first writing $h$ as a combination of alignment vectors corresponding to each variable. Using the Count and Phase variables as an example, we decompose $h$ as follows:

$$h = U_{\text{count}}\vec{z}_{\text{count}} + U_{\text{phase}}\vec{z}_{\text{phase}} + U_{extra}\vec{z}_{extra} - b \tag{10}$$

where $\vec{z}_{\text{var}} \in R^{d_{\text{var}}}$ and $U_{\text{var}} \in R^{d_m \times d_{\text{var}}}$. We can now solve for $\vec{z}_{count}$ by rearranging Equation 10:

$$U_{count}\vec{z}_{count} = h - U_{phase}\vec{z}_{phase} - U_{extra}\vec{z}_{extra} + b \tag{11}$$

In the case where $U$ is orthogonal, then $U_{count}^{-1} = U_{count}^{\top}$ and $U_{count}^{\top}U_{phase} = U_{count}^{\top}U_{extra} = 0$, meaning that $\vec{z}_{count}$ must be defined independently of $\vec{z}_{phase}$. This is not the case, however, for a non-orthogonal $U$, which means that each $\vec{z}_{var}$ can be written in linear terms of the other variables in the SA when using LAFs.

### 4.1.6 Visualization of symbolic algorithm variables with alignment functions

While IIA alone may not be fully diagnostic of the SA that best characterizes the computations performed by an NN, further analysis based on a successful alignment can potentially shed further light on the algorithm a network is using, and it can aid in understanding how neural variables are encoded in the raw neural activity. We provide Figure 4 (a) and (b) to illustrate these points. The Figures were obtained using an LAF with $d_{\text{var}} = 1$ aligned to a Multi-Object LSTM with a hidden dimensionality of 20 ($d_m = 20$). We chose $d_{\text{var}} = 1$ and $d_m = 20$ for visualization purposes, and we use an LAF for its improved IIA when using $d_{\text{var}} = 1$ relative to an OAF on the smaller model size.

Figure 4(a) shows a set of $h$ vectors projected into the 1-dimensional, aligned Phase and Count subspaces taken from the LSTM at many different time steps and trials; we illustrate trajectories in each subspace for individual trials with object counts of 2, 8, and 16. The vertical axis in each panel is used to facilitate the visualization of individual trajectories and their relation to specific token types. One can see in the top panel of (a) how the Phase variable has a value of -1 with slight jitter independent of the demonstration token instance. The Phase then transitions to a positive value at the T token, and stays positive throughout the response phase, though, with some contamination by the Count. Similarly, one can see in the bottom panel of 4(a) how the Count variable progresses from right to left in integer-like steps during the demonstration phase, takes one step further left at the T token, then jumps back 2 steps to the right at the first R token to correctly represent the decremented value of the Count after the first response token. Lastly, it steps rightward until the Count reaches 0, after which the EOS is produced.

Figure 4(b) shows the raw neural activity in the top panel for all 20 of the neurons in the model averaged over 15 sampled trials with object counts of 15, and the bottom two panels show the inverse of *only* the aligned activity for each individual variable, equal to $z_{\text{phase}}u_{\text{phase}} - b$ in the middle panel and $z_{\text{count}}u_{\text{count}} - b$ in the bottom. We provide this figure as a demonstration of how to view the causally relevant neural activity in the NN's original neural space through the lens of the aligned variables. We can see that many of the raw neurons can play a causal role in both the Phase and Count neural coding, confirming notions of distributed coding from prior work (Rumelhart et al., 1986; Smolensky, 1988; Elhage et al., 2022; Olah, 2023).

## 4.2 Transformers

In this section, we demonstrate through empirical and theoretical means that shallow transformers solve the task by recomputing the solution to the task at each step in the sequence. We refer to this class of solutions as anti-Markovian, named for their inductive bias against cumulative, Markovian states. We begin by demonstrating that using the previous transformer layer's outputs as the inputs to the attention mechanism in the next layer restricts transformers from using Markovian solutions that use more steps than attention layers. We then demonstrate a theoretical solution for simplified versions of the Single-Object and Multi-Object numeric equivalence tasks in one layer NoPE transformer architectures, and we causally verify that such a solution emerges empirically. Lastly, we show through causal interventions that similar solutions can emerge in two layer RoPE transformers.

### 4.2.1 Anti-Markovian States

In this section, we demonstrate why Transformer solutions that use Markovian states in the residual stream require a new attention layer for every new step in the sequence. To show this, we focus on a simplified transformer architecture that only includes an embedding layer and the self-attention mechanism within each layer. To justify this simplification, we note that the attention mechanism is the only mechanism in the transformer that provides an opportunity to transmit state information between token positions in the sequence. With this simplification, we can write the output of a single transformer layer as:

$$\begin{bmatrix} h_0^\ell & h_1^\ell & ... & h_t^\ell \end{bmatrix} = \begin{bmatrix} h_0^{\ell-1} & h_1^{\ell-1} & ... & h_t^{\ell-1} \end{bmatrix} + \text{attn}_\ell(\begin{bmatrix} h_0^{\ell-1} & h_1^{\ell-1} & ... & h_t^{\ell-1} \end{bmatrix}) \tag{12}$$

where $h_t^\ell \in R^d$ are column vectors from the transformer residual stream, $\ell$ denotes the attention layer where $\ell = 0$ is the output of the embedding layer, $t$ refers to the positional index in the sequence, and $\text{attn}_\ell(x)$ refers to the attention mechanism. We denote a cumulative state at step $m$ in a Markov chain as $s_m$, and we denote the encoded state in a residual stream vector as $h_t^{\ell,(s_m)}$. We assume that the attn function can only produce and encode $s_{m+1}$ at time $t$ if $s_m$ is already encoded at time $< t$, and we assume that $s_0$ is produced in the embedding layer. Then the cumulative state gets updated with each transformer layer as follows:

$$\begin{bmatrix} h_0^{0,(s_0)} & h_1^0 & h_2^0 & ... & h_t^0 \end{bmatrix} = \text{Embedding}(x_0, x_1, x_2, ..., x_t)$$

$$\begin{bmatrix} h_0^{1,(s_0)} & h_1^{1,(s_1)} & h_2^1 & ... & h_t^1 \end{bmatrix} = \begin{bmatrix} h_0^{0,(s_0)} & h_1^0 & h_2^0 & ... & h_t^0 \end{bmatrix} +$$
$$\text{attn}_1(\begin{bmatrix} h_0^{0,(s_0)} & h_1^0 & h_2^0 & ... & h_t^0 \end{bmatrix})$$

$$\begin{bmatrix} h_0^{2,(s_0)} & h_1^{2,(s_1)} & h_2^{2,(s_2)} & ... & h_t^2 \end{bmatrix} = \begin{bmatrix} h_0^{1,(s_0)} & h_1^{1,(s_1)} & h_2^1 & ... & h_t^1 \end{bmatrix} +$$
$$\text{attn}_2(\begin{bmatrix} h_0^{1,(s_0)} & h_1^{1,(s_1)} & h_2^1 & ... & h_t^1 \end{bmatrix})$$

$$\begin{bmatrix} h_0^{t,(s_0)} & h_1^{t,(s_1)} & h_2^{t,(s_2)} & ... & h_t^{t,(s_t)} \end{bmatrix} = \begin{bmatrix} h_0^{t-1,(s_0)} & h_1^{t-1,(s_1)} & h_2^{t-1,(s_2)} & ... & h_t^{t-1} \end{bmatrix} +$$
$$\text{attn}_t(\begin{bmatrix} h_0^{t-1,(s_0)} & h_1^{t-1,(s_1)} & h_2^{t-1,(s_2)} & ... & h_t^{t-1} \end{bmatrix})$$

Where $x_t$ denotes the input token id at time $t$. We can see that in the best case scenario, the cumulative state can only be transmitted and updated one layer at a time (we provide a more formal proof in Supplement A.7). Thus the two layer transformers in our work are architecturally insufficient for using a solution that involves Markovian states.

We experimentally verify that the two layer RoPE transformers used in this work use anti-Markovian states by performing Transformer Hidden State Substitutions as outlined in Methods Section 3.4.1. Indeed, these substitutions leave the NNs' behavior largely unaffected with an IIA of 0.964 on the original behavior in the Multi-Object RoPE transformers and 0.949 for the Variable-Length Multi-Object RoPE transformers.

We note that generative techniques like scratch pad (Nye et al., 2021) and Chain-of-Thought (CoT) (Wei et al., 2023) allow for transformers to track a cumulative state in the form of self-generated input embeddings. We might expect recurrent models to benefit less from CoT in this respect.

### 4.2.2 Simplified NoPE Transformers

To better understand how a transformer could impement an anti-Markovian solution to the Multi-Object task, we include a theoretical treatment of a single-layer NoPE Transformer that is trained on a simplified version of the Multi-Object task: it excludes the BOS and T tokens from the sequences and only has one possible demo token instance D that is different from the resp token instance R (i.e. an object count of 2 would result in the sequence "D D R R E"). The self-attention calculation for a single query $q_r \in R^d$ from a response token, denoted by the subscript $r$, is as follows:

$$\text{Attention}(q_r, K, V) = V\left(\text{softmax}\left(\frac{K^\top q_r}{\sqrt{d}}\right)\right) = \sum_{i=1}^{n} \frac{e^{\frac{q_r^\top k_i}{\sqrt{d}}}}{\sum_{j=1}^{n} e^{\frac{q_r^\top k_j}{\sqrt{d}}}} v_i = \sum_{i=1}^{n} \frac{s_i^r}{\sum_{j=1}^{n} s_j^r} v_i = \frac{1}{\sum_{j=1}^{n} s_j^r} \sum_{i=1}^{n} s_i^r v_i \tag{13}$$

Where $d$ is the dimensionality of the model, $n$ is the sequence length, $K \in R^{d \times n}$ is a matrix of column vector keys, $V \in R^{d \times n}$ is a matrix of column vectors $v_i$, and $s_i^r = e^{\frac{q_r^\top k_i}{\sqrt{d}}}$, using $i$ to denote the positional index of the key and the superscript $r$ to denote that the $q$ came from a response token. We refer to $s_i^r v_i$ as the strength-value of the $i^{\text{th}}$ token for the query $q_r$.

In the first layer following the embeddings in a NoPE transformer, each of the queries for the response tokens will produce equal strength-values for a given key-value pair regardless of the position from which the response token and demo tokens originated. This is because NoPE does not add positional information to the embeddings. Thus, assuming that the attention mechanism is performing a sum of the count contributions from each token in the sequence, we should be able to use the $s_i^r v_i$ to increment and decrement the model's decision to produce the EOS token from any given response token in the following way:

$$\text{IncrementedAttention}(q_r, K, V) = \frac{1}{s_r^r + \sum_{j=1}^{n} s_j^r}\left(s_r^r v_r + \sum_{i=1}^{n} s_i v_i\right) \tag{14}$$

Where the subscript $r$ in the strength $s_r$ and value $v_r$ denotes that the originating token for the key-value pair is a response token. We can decrement the count using a key-value pair from a demonstration token. To verify our theoretical treatment, we performed a simulation using a single-layer NoPE transformer trained on the simplified Single-Object task. Using the strength-value additions outlined in Equation 14, we were able to change the position at which the transformer produced the EOS token with 100% accuracy. We include results for other transformer architecture variants in Supplemental Figure 6 (c).

### 4.2.3 RoPE Transformers

To determine how the RoPE transformers perform the tasks, we first looked at the attention weights for both of its two layers (see Supplemental Figure 11). The R and EOS queries give surprisingly little attention to the R tokens. In Supplemental Figure 6, we show DAS results on the Input Value variable from the Ctx-Distr SA where a numeric value is assigned to each token and the values of all previous tokens are summed at each step in the sequence. The Multi-Object transformers achieved an IIA of 0.800 for this alignment. We took this to mean that the Multi-Object transformers were at least partially using a positional readout to solve the task. To understand transformer solutions that rely less on a positional readout, we also examined a set of transformers trained on the Variable-Length variant of the Multi-Object task that disrupts count-position correlations. These transformers achieved a higher IIA of 0.935 for the same DAS alignment. The lower IIA of the original Multi-Object transformers is consistent with the notion that they rely, in part, on a positional readout, rather than a summing operation, to solve the task.

## 5 Conclusion

In this work we used both causal and correlational methods to interpret emergent representations of numbers in several types of NNs. We used these methods to discover and characterize graded, symbol-like number variables within the representations of two types of RNNs; we introduced an extension of DAS that adds flexibility our ability characterize neural activity in terms of functions of interpretable symbolic variables; we explored theoretical and empirical transformer solutions to the tasks; and we showed that transformers must use anti-Markovian solutions in the absence of sufficient layers.

As we have noted before in this article, ANNs are by definition symbolic programs, but the algorithms they learn are latent in their connection weights and are not directly observable by inspection of the ANN code. Our goal in aligning NNs and SAs is to find predictive, interpretable, and controllable ways of understanding the learned neural activity. Because NNs are example-driven, gradient-based learners, their learned solutions do not necessarily need to match interpretable SAs exactly. Under idealized conditions, they can come to conform more and more precisely to such algorithms, but they generally do so gradually through training. We believe these points characterize many aspects of the algorithms humans and animals learn through experience, making NNs good models of many aspects of human cognition and development.

When networks learn to conform to simple SAs, they tend to do so in narrow task settings, disconnected from the richness and complexity of real experience Servan-Schreiber et al. (1991). This situation applies to

our models, whose experience is extremely narrow, limited to correct instances of a sequence-based numeric equivalence task. In more complex cases, such as natural language, debate has raged for decades about whether NNs are adequate for extracting the symbolic rules that many argue underlie human language abilities. In our view McClelland (2015), such rules are often useful characterizations for the goals of interpretability, but should not be embraced as the complete story without skepticism. Nevertheless, we strongly encourage work that seeks to understand the computations of neural systems through their alignment with SAs.

## 6 Acknowledgments

Thank you to the PDP Lab for thoughtful discussion and the Stanford Psychology department for funding. Thank you to the ICLR and TMLR reviewers for greatly improving the paper. And huge thank you to Zen Wu for great discussions and assisting with numerous questions about DAS early in the project's development.

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

# A Appendix / supplemental material

## A.1 Additional Figures

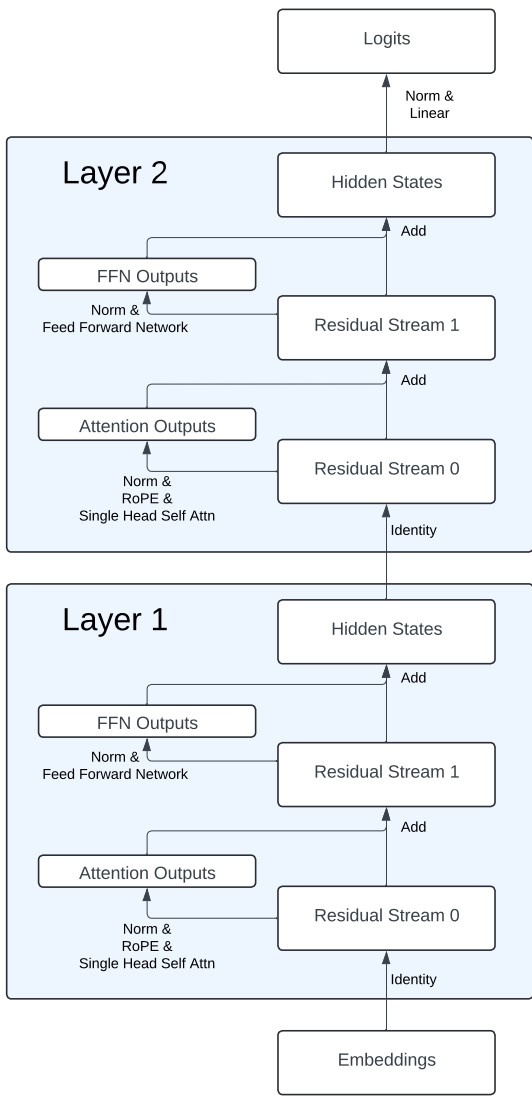

Figure 5: Diagram of the main transformer architecture used in this work. The white rectangles represent activation vectors. The arrows represent model operations. All normalizations are Layer Norms (Ba et al., 2016). The majority of the DAS interchange interventions are performed on Hidden State activation vectors from Layer 1 at individual time-steps. We offer further granularity in the Input Value interventions by performing DAS on the embeddings that are projected into the key and value vectors for the Layer 1 self-attention.

Table 1: The DAS results for each model and task variant. Each alignment function was trained on a single causal variable with a $d_{\text{var}}$ (subspace size) of the better performing out of 16 or 64 dimensions. Performance values are reported as IIA under the task name.

| Model | Alignment | Algorithm | Variable | Multi-Object | Same-Object | Single-Object |
|-------|-----------|-----------|----------|--------------|-------------|---------------|
| GRU | Orthogonal | Up, Down | Count | 0.9464 | 0.3698 | 0.908 |
| GRU | Orthogonal | Up, Down | Phase | 0.9368 | 0.373 | 0.889 |
| GRU | Orthogonal | Up, Up | Demo Count | 0.4807 | 0.2503 | 0.606 |
| GRU | Orthogonal | Up, Up | Resp Count | 0.4173 | 0.4024 | 0.477 |
| GRU | Linear | Up, Down | Count | 0.9906 | 0.991 | 0.995 |
| GRU | Linear | Up, Down | Phase | 0.9884 | 0.9922 | 0.991 |
| GRU | Linear | Up, Up | Demo Count | 0.9174 | 0.9744 | 0.984 |
| GRU | Linear | Up, Up | Resp Count | 0.9223 | 0.9808 | 0.989 |
| LSTM | Orthogonal | Up, Down | Count | 0.993 | 0.958 | 0.989 |
| LSTM | Orthogonal | Up, Down | Phase | 0.991 | 0.95 | 0.991 |
| LSTM | Orthogonal | Up, Up | Demo Count | 0.5416 | 0.86 | 0.409 |
| LSTM | Orthogonal | Up, Up | Resp Count | 0.5374 | 0.8007 | 0.439 |
| LSTM | Linear | Up, Down | Count | 0.9928 | 0.9922 | 0.992 |
| LSTM | Linear | Up, Down | Phase | 0.9914 | 0.9912 | 0.99 |
| LSTM | Linear | Up, Up | Demo Count | 0.9846 | 0.9846 | 0.990 |
| LSTM | Linear | Up, Up | Resp Count | 0.9862 | 0.9697 | 0.991 |

Table 2: The DAS results for each model and task variant using a linear alignment function with a $d_{\text{var}} = 1$ (subspace size). Each alignment function was trained on a single causal variable. Performance values are reported as IIA under the task name.

| Model | Alignment | Algorithm | Variable | $d_{\text{var}}$ | Multi-Object | Same-Object |
|-------|-----------|-----------|----------|-------|--------------|-------------|
| GRU | Linear | Count Up, Count Down | Count | 1 | 0.9436 | 0.68 |
| GRU | Linear | Count Up, Count Down | Phase | 1 | 0.8568 | 0.870 |
| GRU | Linear | Count Up, Count Up | Demo Count | 1 | 0.758 | 0.507 |
| GRU | Linear | Count Up, Count Up | Resp Count | 1 | 0.8339 | 0.890 |
| LSTM | Linear | Count Up, Count Down | Count | 1 | 0.9236 | 0.864 |
| LSTM | Linear | Count Up, Count Down | Phase | 1 | 0.955 | 0.86 |
| LSTM | Linear | Count Up, Count Up | Demo Count | 1 | 0.912 | 0.775 |
| LSTM | Linear | Count Up, Count Up | Resp Count | 1 | 0.9494 | 0.900 |

Table 3: The DAS results for the transformers. Each DAS training was performed on a single causal variable with a $d_{\text{var}}$ (subspace size) of the better performing out of 24 or 64 dimensions. Performance values are reported as IIA under the task name.

| Model | Alignment | Algorithm | Variable | Multi-Object | Same-Object |
|-------|-----------|-----------|----------|--------------|-------------|
| NoPE Transformer | Orthogonal | Context Distributed | Input Value | 0.882 | 0.982 |
| NoPE Transformer | Orthogonal | Count Up, Count Down | Count | 0.1112 | 0.110 |
| RoPE Transformer | Orthogonal | Context Distributed | Input Value | 0.8004 | 0.935 |
| RoPE Transformer | Orthogonal | Count Up, Count Down | Count | 0.1274 | 0.124 |

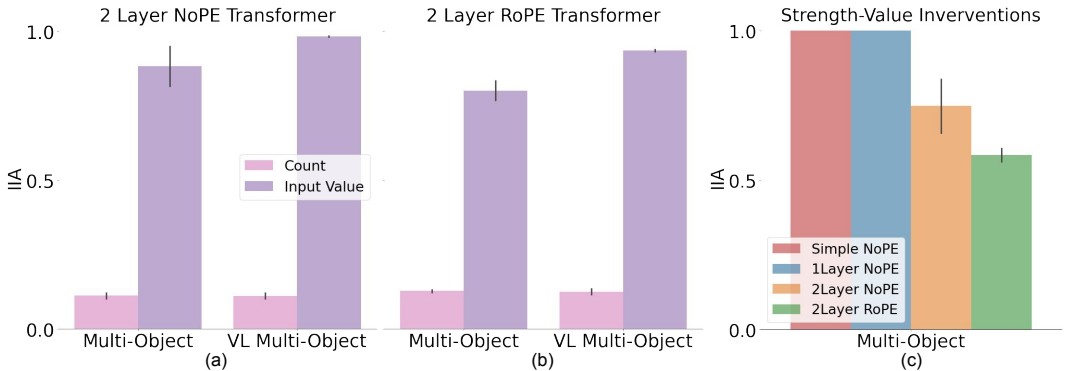

Figure 6: (a) and (b) show the interchange intervention accuracy (IIA) on the Count from the Up-Down program and the Input Value from the Ctx-Distr program aligned to the Transformer architectures using DAS with an Orthogonal alignment function. VL denotes models trained on the Variable-Length version of the task. The Input Value encodes an assigned value (+1, -1, or 0) to each incoming token that is used to recalculate the count at each step in the sequence. The DAS analysis is applied to the model embeddings for the Input Value, and the residual stream after the first transformer layer for the Count. We can see that the Variable-Length transformers have stronger alignment to the Input Value variable—consistent with an interpretation in which the Multi-Object transformers can rely, to some degree, on positional information. (c) IIA for strength-value interventions described in Section 4.2.2. These interventions add and subtract from the count using the strength-value within an attention computation. Strength-values are computed from the last response query, key, and value in the sequence from the layer in which interventions are performed. The displayed IIA is taken from the better performing of the possible attention layers.

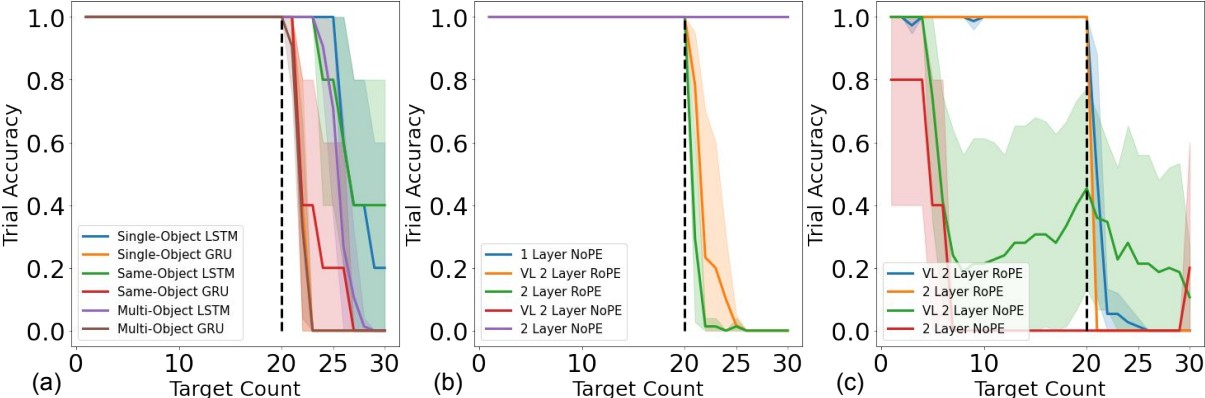

Figure 7: (a) RNN task performance measured as the proportion of trials correct. Object quantity refers to the number of demo tokens in a sequence preceding the trigger token. The evaluation data consists of 15 sampled sequences (even when only one configuration exists for that object quantity). (b) Transformer performance on the Multi-Object task. VL indicates the Variable-Length version of the task. One model seed was dropped from each the NoPE and RoPE models trained on the Variable-Length Multi-Object task due to lower than 99% accuracy. (c) Transformer performance on the Same-Object task. VL indicates the Variable-Length version of the task. All NoPE model seeds performed below 99% accuracy on the Same-Object task.

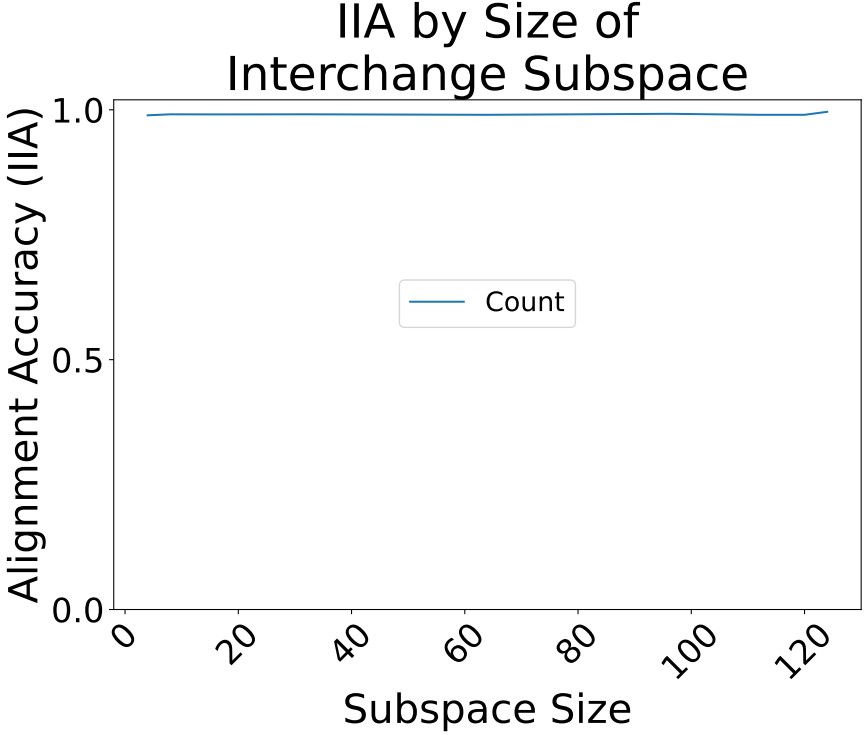

Figure 8: An exploration of the performance of the orthogonal DAS alignment as a function of the size of the interchange subspace for a randomly selected Multi-Object LSTM model seed on the Count variable. The x axis shows $d_{\text{count}}$ while the y axis shows IIA. This is the number of dimensions substituted in the intervention.

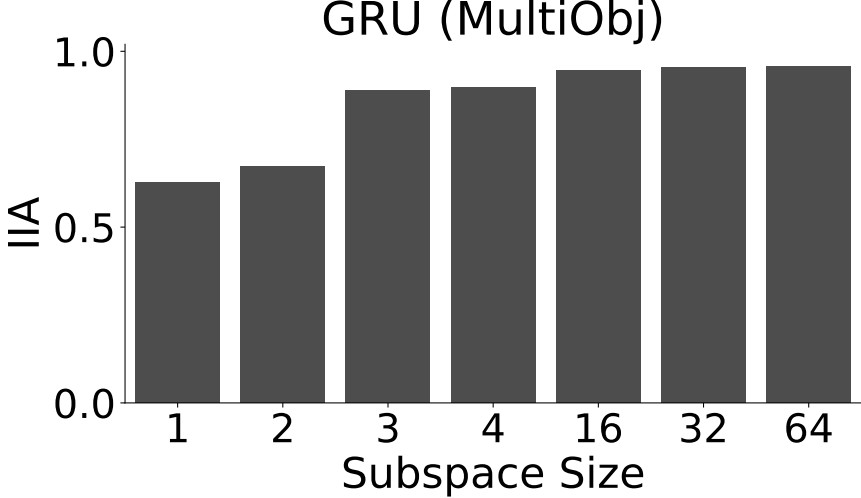

Figure 9: An exploration of the performance of the orthogonal DAS alignment as a function of the size of the interchange subspace for a randomly selected Multi-Object GRU model seed on the Count variable. The x axis shows $d_{\text{count}}$ while the y axis shows IIA. This is the number of dimensions substituted in the intervention.

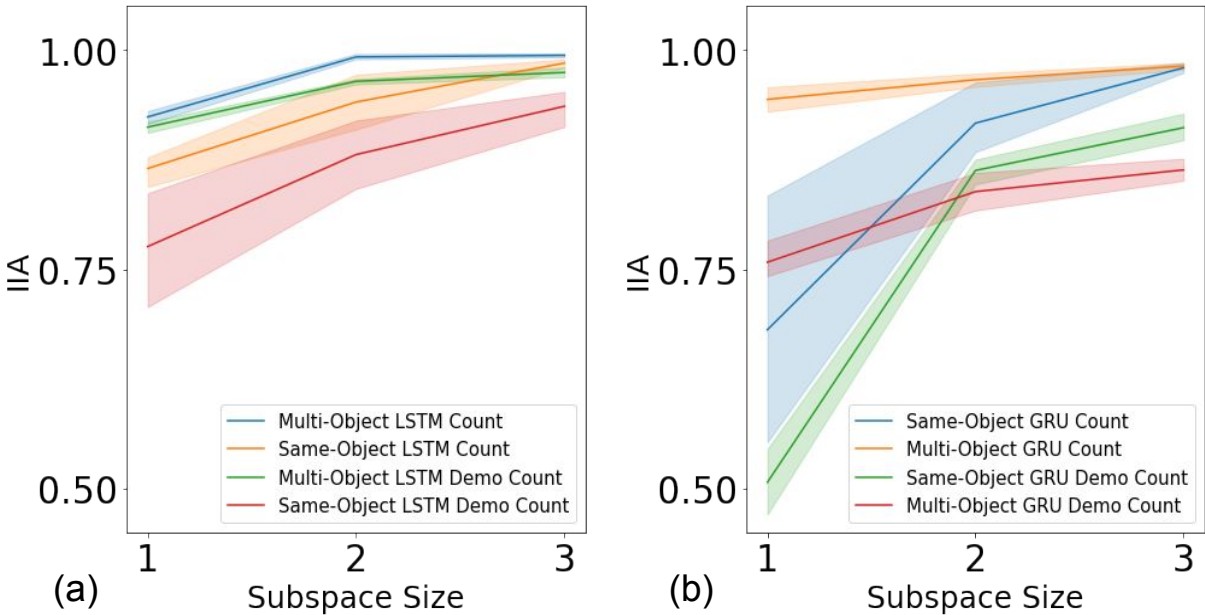

Figure 10: (a) An exploration of the DAS IIA on the y-axis using the Linear Alignment function with varying sizes of $d_{var}$ (the size of the intervention subspace) on the x-axis for the LSTM models. (b) An exploration of the DAS IIA on the y-axis using the Linear Alignment function with varying sizes of $d_{var}$ (the size of the intervention subspace) on the x-axis for the GRU models.

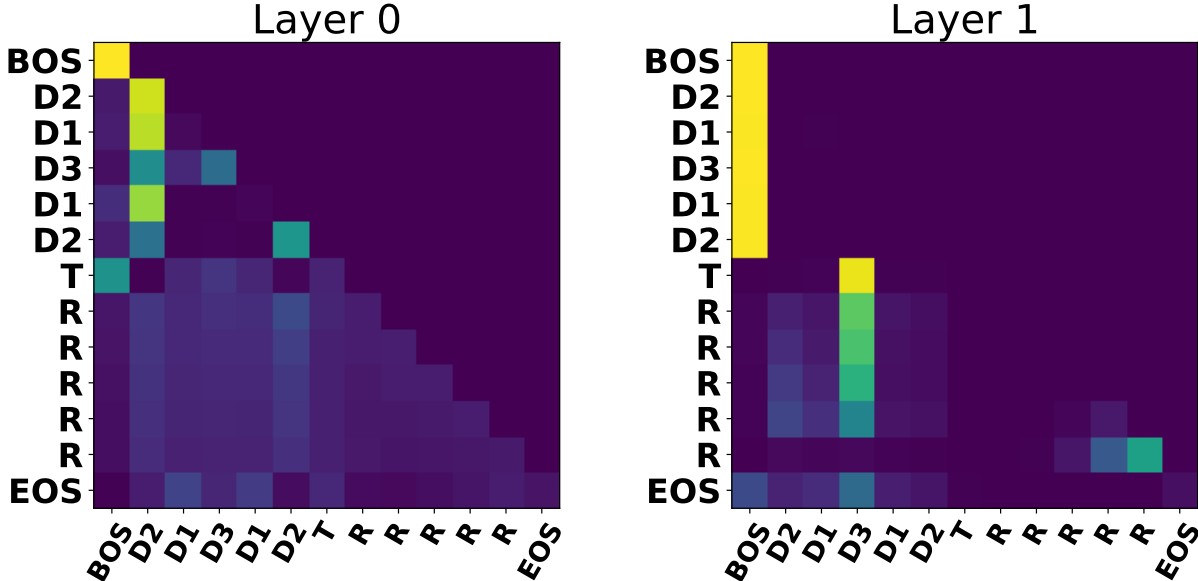

Figure 11: Attention weights for a single transformer with two layers using rotary positional encodings trained on the Multi-Object Task. Queries are displayed on the vertical axis in order of their appearance starting at the top. Keys are displayed on the horizontal axis starting from the left. Queries are only able to attend to themselves and preceding keys.

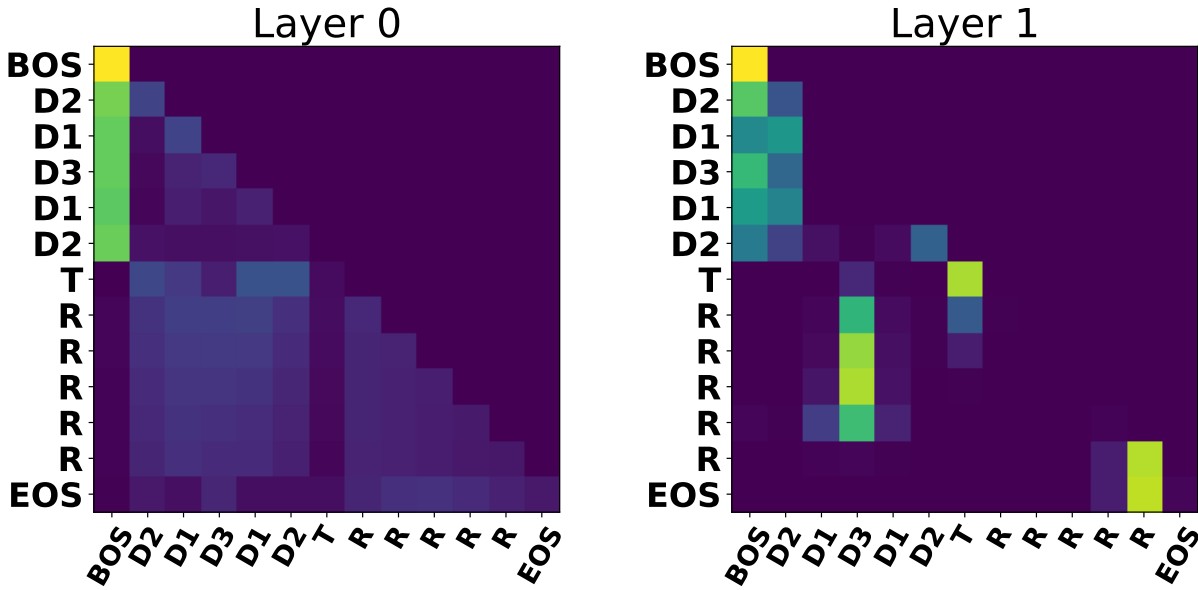

Figure 12: Attention weights for a single transformer with two layers using rotary positional encodings trained on the Variable-Length variant of the Multi-Object Task. Queries are displayed on the vertical axis in order of their appearance starting at the top. Keys are displayed on the horizontal axis starting from the left. Queries are only able to attend to themselves and preceding keys.

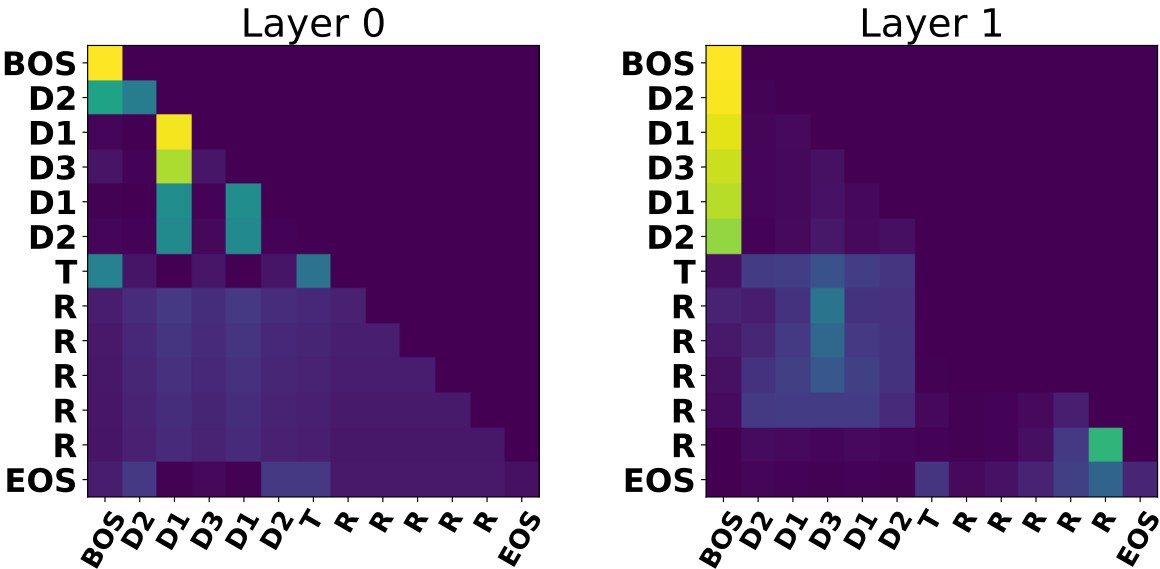

Figure 13: Attention weights for a single transformer model seed with two layers and no positional encodings (NoPE) trained on the Multi-Object Task. Queries are displayed on the vertical axis in order of their appearance starting at the top. Keys are displayed on the horizontal axis starting from the left. Queries are only able to attend to themselves and preceding keys.

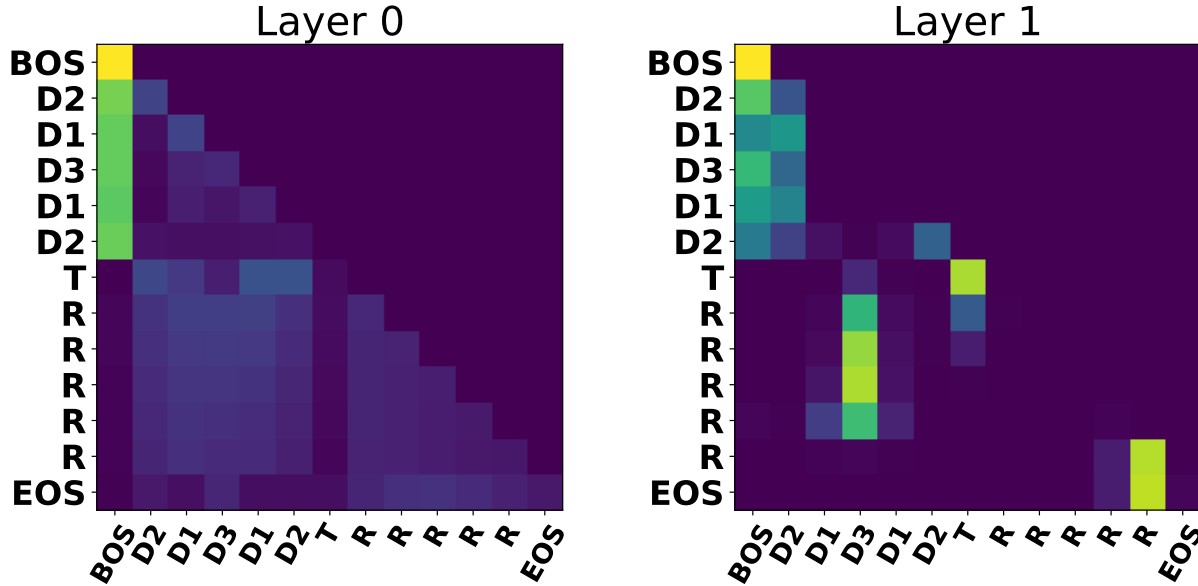

Figure 14: Attention weights for a single transformer with two layers using no positional encodings (NoPE) trained on the Variable-Length variant of the Multi-Object Task. Queries are displayed on the vertical axis in order of their appearance starting at the top. Keys are displayed on the horizontal axis starting from the left. Queries are only able to attend to themselves and preceding keys.

## A.2 Model Details

All artificial neural network models were implemented and trained using PyTorch (Paszke et al., 2019) on Nvidia Titan X GPUs. Unless otherwise stated, all models used an embedding and hidden state size of 128 dimensions. To make the token predictions, each model used a two layer multi-layer perceptron (MLP) with GELU nonlinearities, with a hidden layer size of 4 times the hidden state dimensionality with 50% dropout on the hidden layer. The GRU and LSTM model variants each consisted of a single recurrent cell followed by the output MLP. Unless otherwise stated, the transformer architecture consisted of two layers using Rotary positional encodings (Su et al., 2023). Each model variant used the same learning rate scheduler, which consisted of the original transformer (Vaswani et al., 2017) scheduling of warmup followed by decay. We used 100 warmup steps, a maximum learning rate of 0.0001 , a minimum of 1e-7, and a decay rate of 0.5. We used a batch size of 128, which caused each epoch to consist of 8 gradient update steps.

## A.3 Increment-Up Symbolic Algorithm

In this section we explore an additional SA that we name the **Increment-Up Program**. This SA was in part inspired by the PCA shown in Figure 17. This SA has 3 variables: the Phase, Progress, and the Increment. The algorithm uses the Progress variable to track progress along a fixed interval from 0 to the maximum count which is a preset constant. We refer to the maximum count as the Interval. To solve the numeric equivalence tasks, the algorithm increments the Progress according to an increment size which is defined by the Increment variable. During the demonstration phase, the algorithm initializes the Increment to $\frac{1}{Interval}$. In our case the Interval constant is 20, but we refer to it as the Interval variable for generality. The algorithm progressively adds this increment multiplied by the Interval to the Progress with each successive demonstration token: $Progress = Progress + Increment * Interval$. Upon reaching the trigger token at the end of the demonstration phase, the Increment is set to $\frac{1}{Progress}$ and Progress is subsequently reset to 0. The algorithm then increments the Progress by the new value of Increment (still multiplied by the Interval) until the Progress reaches a value greater than or equal to the Interval. At this point, the SA returns the EOS token. See Algorithm 4 for more details.

We compare the Increment-Up and Up-Down alignments to Multi-Object and Same-Object GRUs in Figure 15. This figure was made from the same model seed used to produce Figure 17. We see that IIA for the Increment-Up SA is relatively low for all considered models in the OAF case. IIA is better in the LAF case, but still does not achieve the strong accuracy of the Up-Down SA. We use this result to demonstrate that LAFs cannot find good alignments with all SAs.

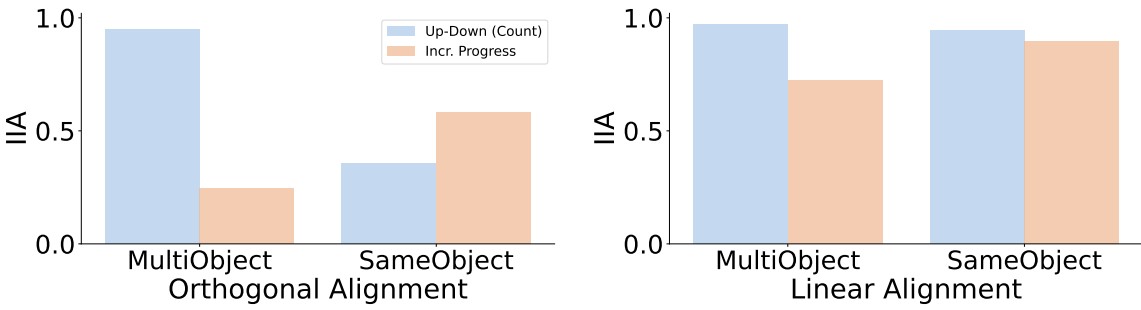

Figure 15: The Interchange intervention accuracy (IIA) for variables from the Up-Down and Increment-Up SAs using different alignment functions for a single model seed for each of the Multi-Object and Same-Object GRUs. The model seed is the same as that used for Figure 17. The displayed IIA is for the Count variable from the Up-Down SA (Up-Down Count) and the Progress variable from the Increment-Up SA (Incr. Progress). We see that even though the LAF raises IIA for both SAs, it does not achieve comparable accuracies. We use the Increment-Up SA as a negative baseline experiment to demonstrate the non-triviality of high LAF IIA.

## A.4 DAS Training Details

### A.4.1 Rotation Matrix Training

To train the DAS rotation matrices, we applied PyTorch's default orthogonal parametrization to a square matrix of the same size as the model's state dimensionality. PyTorch creates the orthogonal matrix as the exponential of a skew symmetric matrix. In all experiments, we selected the number of dimensions to intervene upon as half of the dimensionality of the state. We chose this value after an initial hyperparameter search that showed the number of dimensions had little impact on performance (see Figure 8). We sample 10,000 sequence pairs for the intervention training dataset. See Supplement A.4.3 for more details on intervention data construction and examples. We use a learning rate of 0.001 and a batch size of 512. We removed models with performance below 99% to limit our DAS results to perfectly performing models thus simplifying our interpretations of the results. We chose 99% accuracy instead of 100% due to slight numerical underflow in accuracy calculations and due the fact that half of the Variable-Length Same-Object models would have been dropped due to low performance.

### A.4.2 Symbolic Program Algorithms

---

**Algorithm 1** One sequence step of the Up-Down Program

---

$q \leftarrow$ Count
$p \leftarrow$ Phase
$y \leftarrow$ input token
**if** $y ==$ BOS **then**             ▷ BOS is beginning of sequence token
     $q \leftarrow 0, p \leftarrow 0$
     return sample(D)            ▷ sample a demo token
**else if** $y \in$ D **then**             ▷ D is set of demo tokens
     $q \leftarrow q + 1$
     return sample(D)
**else if** $y ==$ T **then**             ▷ T is trigger token
     $p \leftarrow 1$
**else if** $y ==$ R **then**             ▷ R is response token
     $q \leftarrow q - 1$
**end if**
**if** $(q == 0) \, \& \, (p == 1)$ **then**
     return EOS            ▷ EOS is end of sequence token
**end if**
return R

---

---

**Algorithm 2** One sequence step of the Up-Up Program

---

$d \leftarrow$ Demo Count
$r \leftarrow$ Resp Count
$p \leftarrow$ Phase
$y \leftarrow$ input token
**if** $y ==$ BOS **then**                                                                    ▷ BOS is beginning of sequence token
    $d \leftarrow 0, r \leftarrow 0, p \leftarrow 0$
    return sample(D)                                                        ▷ sample a demo token
**else if** $y \in$ D **then**                                                              ▷ D is set of demo tokens
    $d \leftarrow d + 1$
    return sample(D)
**else if** $y ==$ T **then**                                                               ▷ T is trigger token
    $p \leftarrow 1$
**else if** $y ==$ R **then**                                                               ▷ R is response token
    $r \leftarrow r + 1$
**end if**
**if** $(d == r)$ & $(p == 1)$ **then**
    return EOS                                                               ▷ EOS is end of sequence token
**end if**
return R

---

---

**Algorithm 3** One sequence step of the specific Ctx-Distr Program

---

$v \leftarrow$ list of previous values excluding the most recent step
$\ell \leftarrow$ Input Value                                                               ▷ The value of the most recent token
$p \leftarrow$ Phase                                                  ▷ 0 indicates the demo phase, 1 is the response phase
$y \leftarrow$ input token

$v$.append($\ell$)
$s \leftarrow$ SUM($v$)
**if** $y ==$ BOS **then**                                                                  ▷ BOS is beginning of sequence token
    $\ell \leftarrow 0, p \leftarrow 0$
    return sample(D)                                                        ▷ sample a demo token
**else if** $s \leq 0$ and $p == 1$ **then**                                ▷ Sum is 0 or less in the response phase
    return EOS                                                               ▷ EOS is end of sequence token
**else if** $y ==$ T or $y ==$ R **then**                             ▷ T is trigger token, R is response token
    $p \leftarrow 1$
    $\ell \leftarrow -1$
    return R
**else if** $y \in$ D **then**                                                              ▷ D is set of demo tokens
    $\ell \leftarrow 1$
**end if**

**if** $p == 1$ **then**
    return R
**else**
    return sample(D)
**end if**

---

---

**Algorithm 4** One sequence step of the Increment-Up Program

---

$m \leftarrow$ Interval
$q \leftarrow$ Progress
$p \leftarrow$ Phase
$i \leftarrow$ Increment
$y \leftarrow$ input token
**if** $y ==$ BOS **then**                                                                 ▷ BOS is beginning of sequence token
    $q \leftarrow 0, p \leftarrow 0, i \leftarrow \frac{1}{m}$
    return sample(D)                                                       ▷ sample a demo token
**else if** $(y \in$ D or $y ==$ R$)$ and $q < m$ **then**
    $q \leftarrow q + i * m$
**else if** $y ==$ T **then**                                                            ▷ T is trigger token
    $p \leftarrow 1$
    $i \leftarrow \frac{1}{q}$
    $q \leftarrow 0$
**end if**
**if** $(q \geq m)$ and $(p == 1)$ **then**
    return EOS                                                              ▷ EOS is end of sequence token
**else if** $q \geq m$ and $p == 0$ **then**
    return T
**else if** $p == 0$ **then**
    return sample(D)
**else**
    return R
**end if**

---

### A.4.3 DAS Intervention Data

Here we expand upon the intervention data used to train and test the DAS rotation matrices. We organize this section into programs, variables, and tasks. For each DAS training, we train a single orthonormal matrix and only create interventions that depend on a single variable from the corresponding program. To construct an intervention sample, we first sample a target sequence and a source squence and a positional index from each sequence. We limit positional indices to the demo and resp tokens. We then compute the values of each of the variables using the symbolic algorithm up to the positional index for both the target and source. The value of the variable of focus is then transferred from the source into the the target variable. We then continue the target sequence based on the new value. When the target sequence's counterfactual sequence begins in the demo phase, we uniformly sample the number of demo sequence steps before placing the trigger token such that the Count (or Demo Count ) does not exceed the maximum count used in the task. We note that this makes the samples not strictly counterfactual in the definition used in the causal inference literature, but the desired effect is the same as the true counterfactual comes from the same distribution.

### A.4.4 Up-Down Program Examples

**Count Variable:** Interventions attempt to transfer the representation corresponding to the difference between the number of resp tokens and demo tokens. Interventions are only performed at positional indices corresponding to demo or resp tokens.

| Multi-Object Examples | 1 | 2 | 3 | 4 |
|---|---|---|---|---|
| Source Sequence | BOS $D_1$ | BOS $D_2$ $D_1$ $D_1$ | BOS $D_2$ $D_1$ T R | BOS $D_1$ $D_3$ T R R |
| Target Sequence | BOS $D_3$ $D_2$ | BOS $D_2$ T R | BOS $D_1$ $D_2$ $D_1$ T R | BOS $D_2$ |
| Original Labels | $D_2$ $D_3$ T R R R R EOS | EOS | R R EOS | $D_2$ T R R EOS |
| Counterfactual | $D_2$ $D_3$ T R R R EOS | R R R EOS | R EOS | $D_2$ T R EOS |

| Single-Object Examples | 1 | 2 | 3 | 4 |
|---|---|---|---|---|
| Source Sequence | BOS D | BOS D D D | BOS D D T R | BOS D D T R R |
| Target Sequence | BOS D D | BOS D T R | BOS D D D T R | BOS D |
| Original Labels | D D T R R R R EOS | EOS | R R EOS | D T R R EOS |
| Counterfactual | D D T R R R EOS | R R R EOS | R EOS | D T R EOS |

| Same-Object Examples | 1 | 2 | 3 | 4 |
|---|---|---|---|---|
| Source Sequence | BOS C | BOS C C C | BOS C C T C | BOS C C T C C |
| Target Sequence | BOS C C | BOS C T C | BOS C C C T C | BOS C |
| Original Labels | C C T C C C C EOS | EOS | C C EOS | C T C C EOS |
| Counterfactual | C C T C C C EOS | C C C EOS | C EOS | C T C EOS |

**Phase Variable:** Interventions transfer the representation corresponding to the Phase of the sequence (whether it is counting up or counting down). Interventions are only performed at positional indices corresponding to demo or resp tokens.

| Multi-Object Examples | 1 | 2 | 3 | 4 |
|---|---|---|---|---|
| Source Sequence | BOS $D_1$ | BOS $D_3$ $D_1$ $D_2$ | BOS $D_2$ $D_1$ T R | BOS $D_2$ $D_3$ T R R |
| Target Sequence | BOS $D_2$ $D_1$ | BOS $D_3$ T R | BOS $D_1$ $D_3$ $D_1$ T R | BOS $D_2$ |
| Original Labels | $D_3$ $D_1$ T R R R R EOS | EOS | R R EOS | $D_1$ T R R EOS |
| Counterfactual | $D_3$ $D_1$ T R R R R EOS | $D_2$ T R EOS | R R EOS | R EOS |

| Single-Object Examples | 1 | 2 | 3 | 4 |
|---|---|---|---|---|
| Source Sequence | BOS D | BOS D D D | BOS D D T R | BOS D D T R R |
| Target Sequence | BOS D D | BOS D T R | BOS D D D T R | BOS D |
| Original Labels | D D T R R R R EOS | EOS | R R EOS | D T R R EOS |
| Counterfactual | D D T R R R R EOS | D T R EOS | R R EOS | R EOS |

| Same-Object Examples | 1 | 2 | 3 | 4 |
|---|---|---|---|---|
| Source Sequence | BOS C | BOS C C C | BOS C C T C | BOS C C T C C |
| Target Sequence | BOS C C | BOS C T C | BOS C C C T C | BOS C |
| Original Labels | C C T C C C C EOS | EOS | C C EOS | C T C C EOS |
| Counterfactual | C C T C C C C EOS | C T C EOS | C C EOS | C EOS |

### A.4.5 Up-Up Program Examples

**Demo Count Variable:** Interventions attempt to transfer the representation corresponding to the number of demo tokens in the sequence. Interventions are only performed at positional indices corresponding to demo or resp tokens. We remove training and evaluation samples in which the Demo Count is less than the Resp Count .

| *Multi-Object Examples* | 1 | 2 | 3 | 4 |
|---|---|---|---|---|
| Source Sequence | BOS $D_1$ | BOS $D_2$ $D_3$ $D_3$ | BOS $D_2$ $D_1$ T R R | BOS $D_1$ $D_3$ T R R |
| Target Sequence | BOS $D_3$ $D_2$ | BOS $D_2$ $D_2$ $D_3$ T R R | BOS $D_1$ $D_2$ $D_1$ T R | BOS $D_2$ |
| Original Labels | T R R EOS | R EOS | R R EOS | $D_2$ T R R EOS |
| Counterfactual | T R EOS | R EOS | R EOS | $D_2$ T R R R EOS |

| *Single-Object Examples* | 1 | 2 | 3 | 4 |
|---|---|---|---|---|
| Source Sequence | BOS D | BOS D D D | BOS D D T R R | BOS D D T R R |
| Target Sequence | BOS D D | BOS D D D T R R | BOS D D D T R | BOS D |
| Original Labels | T R R EOS | R EOS | R R EOS | D T R R EOS |
| Counterfactual | T R EOS | R EOS | R EOS | D T R R R EOS |

| *Same-Object Examples* | 1 | 2 | 3 | 4 |
|---|---|---|---|---|
| Source Sequence | BOS C | BOS C C C | BOS C C T C C | BOS C C T C C |
| Target Sequence | BOS C C | BOS C C C T C C | BOS C C C T C | BOS C |
| Original Labels | T C C EOS | C EOS | C C EOS | C T C C EOS |
| Counterfactual | T C EOS | C EOS | C EOS | C T C C C EOS |

**Resp Count Variable:** Interventions attempt to transfer the representation corresponding to the number of response tokens in the sequence. Interventions are only performed at positional indices corresponding to demo or resp tokens. We remove samples from the training and evaluation sets that transfer a Resp Count greater than the Demo Count into the response phase.

| *Multi-Object Examples* | 1 | 2 | 3 | 4 |
|---|---|---|---|---|
| Source Sequence | BOS $D_1$ $D_3$ $D_3$ | BOS $D_2$ | BOS $D_2$ $D_1$ T R R | BOS $D_1$ $D_3$ $D_3$ T R R R |
| Target Sequence | BOS $D_3$ $D_2$ | BOS $D_2$ $D_2$ $D_3$ T R R | BOS $D_1$ $D_2$ $D_1$ T R | BOS $D_2$ |
| Original Labels | T R R EOS | R EOS | R R EOS | $D_2$ T R R EOS |
| Counterfactual | T R R EOS | R R R EOS | R EOS | $D_2$ T EOS |

| *Single-Object Examples* | 1 | 2 | 3 | 4 |
|---|---|---|---|---|
| Source Sequence | BOS D D D | BOS D | BOS D D T R R | BOS D D D T R R R |
| Target Sequence | BOS D D | BOS D D D T R R | BOS D D D T R | BOS D |
| Original Labels | T R R EOS | R EOS | R R EOS | D T R R EOS |
| Counterfactual | T R R EOS | R R R EOS | R EOS | D T EOS |

| *Same-Object Examples* | 1 | 2 | 3 | 4 |
|---|---|---|---|---|
| Source Sequence | BOS C C C | BOS C | BOS C C T C C | BOS C C C T C C C |
| Target Sequence | BOS C C | BOS C C C T C C | BOS C C C T C | BOS C |
| Original Labels | T C C EOS | C EOS | C C EOS | C T C C EOS |
| Counterfactual | T C C EOS | C C C EOS | C EOS | C T EOS |

### A.4.6 Ctx-Distr Program Examples

**Anti-Markovian States:** We perform these interventions directly by substituting the source hidden state into the target hidden state without using DAS. Each intervention examines whether the state encodes sufficient information to transfer the NN's behavior from the source sequence into the target sequence. If the NN uses a Markovian hidden state, then transferring the hidden state from one position to another should result in a corresponding transfer of behavior. In the case that the NN uses anti-Markovian states, then we would expect the model's behavior to be unchanged at token positions that did not receive interventions. Higher accuracies correspond to no behavioral transfer. Interventions are only performed at positional indices corresponding to non-terminal response tokens.

| *Multi-Object Examples* | 1 | 2 | 3 | 4 |
|---|---|---|---|---|
| Source Sequence | BOS $D_1$ $D_3$ T R R | BOS $D_2$ T R | BOS $D_2$ $D_1$ T R | BOS $D_1$ $D_3$ $D_3$ T R R |
| Target Sequence | BOS $D_3$ $D_2$ T R | BOS $D_2$ $D_2$ $D_3$ T R | BOS $D_1$ $D_2$ T R R | BOS $D_2$ $D_1$ $D_2$ T R |
| Original Label | R R EOS | R R R EOS | EOS | R R EOS |
| Counterfactual | R R EOS | R R R EOS | EOS | R R EOS |

| *Single-Object Examples* | 1 | 2 | 3 | 4 |
|---|---|---|---|---|
| Source Sequence | BOS D D T R R | BOS DT R | BOS D D D T R | BOS D D D T R R |
| Target Sequence | BOS D D T R | BOS D D D T R | BOS D D T R R | BOS D D D T R |
| Original Label | R R EOS | R R R EOS | EOS | R R EOS |
| Counterfactual | R R EOS | R R R EOS | EOS | R R EOS |

| *Same-Object Examples* | 1 | 2 | 3 | 4 |
|---|---|---|---|---|
| Source Sequence | BOS C C T C C | BOS CT C | BOS C C C T C | BOS C C C T C C |
| Target Sequence | BOS C C T C | BOS C C C T C | BOS C C T C C | BOS C C C T C |
| Original Label | C C EOS | C C C EOS | EOS | C C EOS |
| Counterfactual | C C EOS | C C C EOS | EOS | C C EOS |

**Input Value Variable:** These interventions attempt to transfer the representation corresponding to the value with which the tokens contribute to the cumulative difference between the demo and resp tokens. A value of +1 is assigned to demo tokens, a value of -1 is assigned to resp tokens, and the algorithm stops when the sum of the values is equal to 0 in the resp phase. Interventions are only performed at positional indices corresponding to demo or resp tokens, and we restrict the number of demo tokens to be at least 2 when intervening on the demo phase. This latter restriction is to avoid cases where the cumulative value is negative.

| Multi-Object Examples | 1 | 2 | 3 | 4 |
|---|---|---|---|---|
| Source Sequence | BOS $D_1$ | BOS $D_2$ | BOS $D_2$ $D_1$ T R R | BOS $D_1$ $D_3$ $D_3$ T R R R |
| Target Sequence | BOS $D_3$ $D_2$ | BOS $D_2$ $D_2$ $D_3$ T R R | BOS $D_1$ $D_2$ $D_1$ T R | BOS $D_2$ $D_1$ |
| Original Labels | T R R EOS | R EOS | R R EOS | $D_2$ T R R R EOS |
| Counterfactual | T R R EOS | R R R EOS | R R EOS | $D_2$ T R EOS |

| Single-Object Examples | 1 | 2 | 3 | 4 |
|---|---|---|---|---|
| Source Sequence | BOS D | BOS D | BOS D D T R R | BOS D D D T R R R |
| Target Sequence | BOS D D | BOS D D D T R R | BOS D D D T R | BOS D D |
| Original Labels | T R R EOS | R EOS | R R EOS | D T R R R EOS |
| Counterfactual | T R R EOS | R R R EOS | R R EOS | D T R EOS |

| Same-Object Examples | 1 | 2 | 3 | 4 |
|---|---|---|---|---|
| Source Sequence | BOS C | BOS C | BOS C C T C C | BOS C C C T C C C |
| Target Sequence | BOS C C | BOS C C C T C C | BOS C C C T C | BOS C C |
| Original Labels | T C C EOS | C EOS | C C EOS | C T C C C EOS |
| Counterfactual | T C C EOS | C C C EOS | C C EOS | C T C EOS |

### A.4.7  DAS Gradience Evaluation Data

The data used for Figures 3 (c) and (d) was constructed by sampling a single target sequence for every object count ranging from 1-20, and source sequences with object counts incrementing by 4 for each target sequence. Interventions were then constructed for each target count, source count pair within each sequence pair. This procedure was repeated for three times, each with a different number of steps in the demo phase before providing the trigger token. The number of continued demo steps was 1, 4, and 12 respectively.

### A.5  Context Distributed Interventions

We detail in this section why our Ctx-Distr interchange interventions are sufficient to demonstrate that the transformers use a solution that re-references/recomputes the relevant information to solve the tasks at each step in the sequence. The hidden states in Layer 1 are a bottleneck at which a cumulative counting variable must exist if it were to use a strategy like the Up-Down or Up-Up programs. This is because the Attention Outputs of Layer 1 are the first activations that have had an opportunity to communicate across token positions. This means that the representations between the Residual Stream 1 of Layer 1 up to the Residual Stream 0 of Layer 2 cannot have read a cumulative state from the previous token position other than reading off the positional information from the previous positional encodings. The 2-layer architecture is then limited in that it has only one more opportunity to transfer information between positions—the attention mechanism in Layer 2. Thus, if a hidden state at time $t$ were to have encoded a cumulative representation of the count that will be used by the model at time $t + 1$, that cumulative representation must exist in the activation vectors between the Residual Stream 1 in Layer 1 and the Residual Stream 0 of Layer 2. If it is using such a cumulative representation, then when we perform a full activation swap in the Layer 1 hidden states then the resulting predictions should be influenced by the swap.

### A.6  Variable-Length Task Variants

Here we include additional tasks to prevent the transformers with positional encodings from learning a solution that relies on reading out positional information. We introduce Variable-Length variants of each of the Multi-Object, Single-Object, and Same-Object tasks. In the Variable-Length versions, each token in the demo phase has a 0.2 probability of being sampled as a unique "void" token type, V, that should be ignored when determining the object quantity of the sequence. The number of demo tokens will still be equal to the object quantity when the trigger token is presented. We include these void tokens as a way to vary the length of the demo phase for a given object quantity, thus breaking correlations between positional information and object quantities. As an example, consider the possible sequence with a object quantity of 2: "BOS V D V V D T R R EOS".

### A.7 Anti-Markovian Proof

**Notation.** We use the following symbols throughout the theorem and proof:

- $x_i$: the input token ID at position $i$ in the sequence.

- $\mathrm{Embed}(x_i)$: the embedding of token $x_i$; this is taken to be the "residual stream" output of layer $\ell = 0$.

- $\ell$: the layer index. We number layers so that $\ell = 0$ is the embedding layer, and $\ell = 1, 2, \ldots$ are the successive self-attention layers.

- $t$: the final position in the sequence whose cumulative state $s_t$ we wish to encode.

- $h_i^\ell \in \mathbb{R}^d$: the residual-stream vector at layer $\ell$ and position $i$. In particular,

$$h_i^0 = \begin{cases} s_0, & i = 0, \\ \mathrm{Embed}(x_i), & i \geq 1, \end{cases}$$

  and for $\ell \geq 1$,

$$h_i^\ell = h_i^{\ell-1} + \mathrm{attn}_\ell\big(h_0^{\ell-1}, \ldots, h_i^{\ell-1}\big).$$

- $\mathrm{attn}_\ell(\cdot)$: the causal self-attention update at layer $\ell$, which may attend only to token-wise linear functions of positions $\leq i$ when computing $h_i^\ell$.

- $s_i$: the "Markovian" cumulative state after seeing tokens up to position $i$. By hypothesis,

$$s_i = g\big(s_{i-1}, x_i\big), \quad s_0 = g(\cdot, x_0),$$

  and we require $h_i^\ell = s_i$ exactly when $\ell \geq i$.

- $g$: the state-update function $g$: (previous state, current token) $\mapsto$ new state.

- #layers: the total number of self-attention layers in the Transformer (excluding the embedding layer).

**Theorem:** In a Transformer with causal self-attention, suppose that after $\ell$ layers, position $i$ in the residual stream carries the full Markovian state

$$s_i = g(s_{i-1}, x_i)$$

if and only if $\ell \geq i$. Then to encode $s_t$ at position $t$ one must have

$$\#\mathrm{layers} \geq t.$$

We proceed by induction on the number of layers $\ell$.

**Base case** ($\ell = 0$). Layer 0 is just the embedding layer:

$$h_0^0 = s_0 = g(\cdot, x_0), \quad h_i^0 = \mathrm{Embed}(x_i) \quad (i \geq 1).$$

Thus only position 0 carries the cumulative state, and for any $t \geq 1$, $\ell = 0 < t$ is insufficient to encode $s_t$.

**Inductive step**. Assume that after $\ell$ layers,

$$\begin{cases} h_i^\ell = s_i, & i \leq \ell, \\ h_i^\ell \neq s_i, & i > \ell. \end{cases}$$

Consider layer $\ell + 1$. For each position $i$,

$$h_i^{\ell+1} = h_i^\ell + \mathrm{attn}_{\ell+1}\big(h_0^\ell, \ldots, h_i^\ell\big).$$

- If $i = \ell + 1$, then by causality the attention may attend only to positions $0, \ldots, \ell + 1$. By the inductive hypothesis, for $j \leq \ell$ we have $h_j^\ell = s_j$, and $h_{\ell+1}^\ell = f(x_{\ell+1})$ where $f$ is some function that has not seen or produced $s_{\ell+1}$. Hence the attention head can compute

$$s_{\ell+1} \;=\; g\big(s_\ell, x_{\ell+1}\big)$$

  and add it to its residual stream, yielding $h_{\ell+1}^{\ell+1} = s_{\ell+1}$.

- If $i > \ell + 1$, then no information can traverse more than one new position per layer, so position $i$ still does not have $s_i$.

Therefore after $\ell + 1$ layers exactly positions $0, \ldots, \ell + 1$ carry the states $s_0, \ldots, s_{\ell+1}$. This completes the induction.

Hence to encode the state $s_t$ at position $t$, the Transformer must have at least $t$ layers.

### A.8 Principle Components Analysis

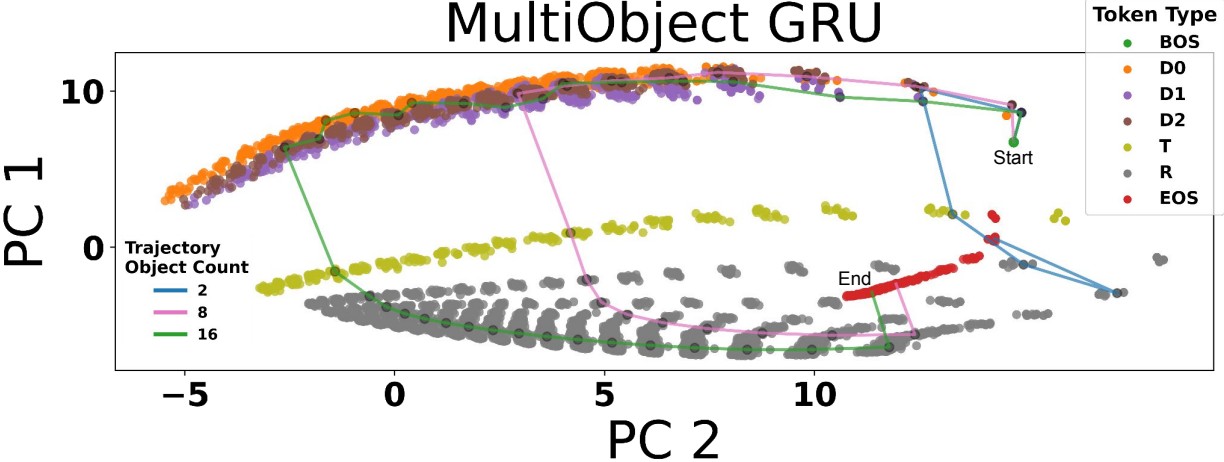

Figure 16: Top 2 Principal Components (PCs) of the hidden states of a single Multi-Object GRU model seed where representations are taken from 15 sampled trials for each object quantity from 1 to 20 in the Multi-Object task variant. Each dot represents a hidden state representation projected into the first 2 PCs. We plot select trial trajectories to exemplify how the states change for a given trial. Green points indicate the start of a plotted trajectory, black points indicate an intermediate step, and red points indicate the end of a plotted trajectory. The blue line plots a single trajectory from start to finish with a object quantity of 2. Similarly, the pink and green lines follow single trajectories of 8 and 16 respectively.

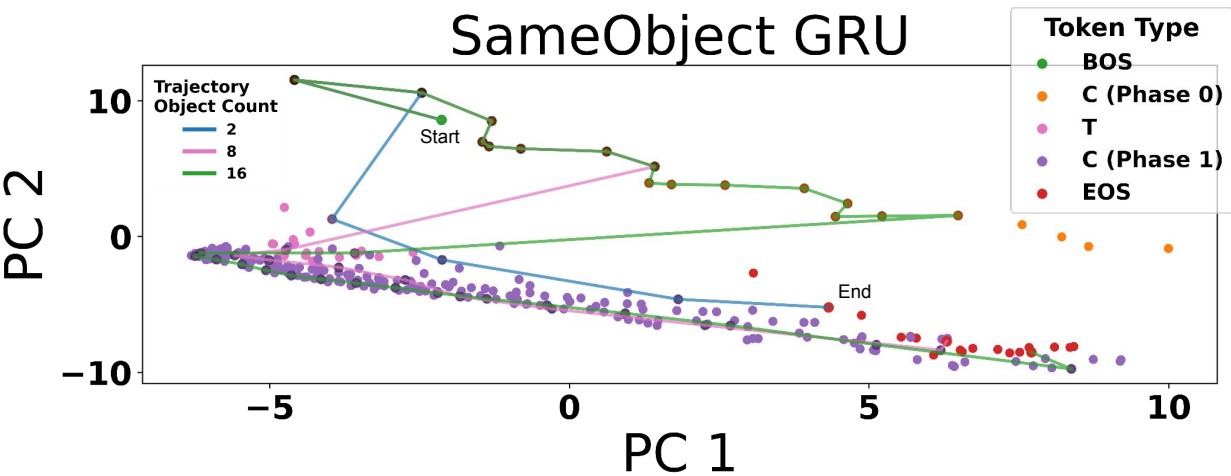

Figure 17: Top 2 Principal Components (PCs) of the hidden states of a single Same-Object GRU model seed where representations are taken from 15 sampled trials for each object quantity from 1 to 20 in the Same-Object task variant. Each dot represents a hidden state representation projected into the first 2 PCs. We plot select trial trajectories to exemplify how the states change for a given trial. Green points indicate the start of a plotted trajectory, black points indicate an intermediate step, and red points indicate the end of a plotted trajectory. The blue line plots a single trajectory from start to finish with a object quantity of 2. Similarly, the pink and green lines follow single trajectories of 8 and 16 respectively.

