# OpenReview forum: "Emergent Symbol-like Number Variables in Artificial Neural Networks"
_TMLR — Accepted by TMLR_

### Review · Reviewer_JpkP · 2025-03-14

**Summary Of Contributions:**

The paper investigates the emergence of symbol-like number representations in neural networks trained on next-token prediction tasks. Specifically, they study GRUs, LSTMs and transformer models, trained on different variations of numeric equivalence tasks. Then, they use distributed alignment search (DAS) to investigate the alignment between the representations and different hypothesised symbolic algorithms.

The key finding is that these models learn symbol-like representations that are used to solve the task. Specifically, they find that recurrent models learn a "count up, count down" algorithm, whereas transformer models adopt a "context-distributed" strategy that queries a history of inputs at each step of the sequence. They also perform additional causal interventions to validate the results of DAS, and study the development of those variables during training.

**Audience:**

Yes

**Broader Impact Concerns:**

I do not have any concerns on the ethical implications of this work.

**Claims And Evidence:**

Yes

**Requested Changes:**

Optional:
- I believe the discussion and conclusion section, which touches on a number of important issues or additional experiments, could benefit from paragraph titles.

**Strengths And Weaknesses:**

Strengths:
- The paper presents an interesting study of how different neural network architectures solve a simple mathematical/symbolic problem.
- The study considers different variations of the problem and different hyperparameters, supporting the robustness of their findings.
- The causal experiments are convincing and support the claims.
- The paper is very well written and easy to follow.

Weaknesses:
- The studied task is relatively simple.
- The fact that neural networks can approximate symbolic computation is not necessarily surprising, given prior work in the field of mechanistic interpretability.

---

> ### Author Response · Authors · 2025-04-14
>
> Thank you for your review, it is helpful. We have broadly reorganized the paper, and have been more diligent about sectioning for clarity. We hope our changes are satisfying. Thank you again for your review, it has helped make the paper more clear.

---

### Review · Reviewer_GPsF · 2025-03-16

**Summary Of Contributions:**

This work examines how RNN and transformer-based language models trained on next-token prediction represent numeric concepts in their internal representations by conducting multiple causal experiments. It provides insights into the differences in the underlying mechanisms that enable RNNs and transformer-based models to perform the same task. Additionally, the study highlights a correlation between the emergence of task performance and the alignment of mechanisms during the training process among other insights.

**Audience:**

Yes

**Broader Impact Concerns:**

I do not have any concerns regarding the ethical implications of this work.

**Claims And Evidence:**

Yes

**Requested Changes:**

**Critical recommendations**:
- For each hypothesized mechanism, clearly enumerate every causal experiment separately, specifying their corresponding counterfactuals and targets—ideally, one for each variable in the mechanism. Explicitly describe which variable in the algorithm is being aligned and explain why the proposed causal experiment is meaningful. Additionally, provide details on the granularity of the patching experiment, clarifying whether it is conducted at the residual stream level, attention subblock level, or individual neuron level.

- Define what is meant by the term “theoretical neural solution,” as mentioned in the second paragraph of Section 4.2 and Figure 3. Furthermore, provide additional context to clarify the reasoning presented in the second paragraph of Section 4.2.

- Lastly, define the term “graded alignment” as introduced in Section 4.3.

**Additional recommendations**:
- Please clarify the rationale for using DAS. While DAS is designed to identify the subspace encoding a specific causal variable, this work does not appear to focus on specific subspaces.

- Additionally, cite [5] and provide an explanation of how its findings could help interpret the results of this study.

[5] Haviv et al, “Transformer Language Models without Positional Encodings Still Learn Positional Information”, 2022.

**Strengths And Weaknesses:**

**Strengths**:
- This work clearly outlines potential hypothesized mechanisms responsible for performing the task and conducts multiple causal experiments to accept or reject these hypotheses.

- It rigorously examines both RNN and transformer-based models, identifying differences in their underlying mechanisms for performing the same task. This analysis provides novel insights into how neural architecture influences the learned mechanisms.

- Unlike most mechanistic interpretability studies, this work not only investigates the mechanisms after training but also explores how they develop during the training process.

**Weakness**:
- My primary concern with this work is the lack of clarity in its experiments, which appear cluttered and do not clearly articulate their purpose. Specifically, it does not explicitly describe which variable(s) of which symbolic algorithm are being aligned. The counterfactuals and targets of each experiment are unclear, making it difficult to understand their rationale and significance.
  - In Section 3.5, the text states that Figure 4 presents results for patching the residual stream after layer 1 to verify the absence of the cumulative count variable—meaning that patching these vectors from different time steps should not affect the final output. However, the first line of Section 4 describes Figure 4 as representing DAS results, creating confusion.
  - Additionally, in Section 3.5, the authors discuss a DAS experiment aimed at aligning the Last Value variable in the Ctx-Distr program but do not specify the counterfactuals and targets. Similarly, patching at the attention subblock is mentioned without clarifying which variable this experiment seeks to align or in which layer(s) it is conducted.
  - Moreover, in Figure 4, the Count variable is aligned for the up-down and up-up SAs, but the corresponding experiment, along with its counterfactuals and target, is not explicitly detailed.

- Finally, this work does not cite several existing studies that have attempted to mechanistically understand how LLMs represent numerical values in their internal activations. Some of these relevant works include [1–4, inter alia].

[1] Kantamneni et al, “Language Models Use Trigonometry to Do Addition”, 2025.

[2] Liu et al, “Towards Understanding Grokking: An Effective Theory of Representation Learning”, 2022.

[3] Nanda et al, “Progress measures for grokking via mechanistic interpretability”, 2023.

[4] Zhong et al, “The Clock and the Pizza: Two Stories in Mechanistic Explanation of Neural Networks”, 2023.

---

> ### Author Response · Authors · 2025-04-14
>
> Thank you for your detailed review, we found it very helpful for directing our edits. We address the issues you raised as follows:
>
> 1. We have provided more detail and examples of intervention data in Supplemental Sections A.3.3-A.3.6. We have also broadly reorganized the paper into RNN and Transformer analyses which will hopefully make the purpose of each analysis more clear. With regards to transformers, we have provided theoretical treatments of the transformers in sections 4.2.1 and 4.2.2 using causal interventions as empirical backing for our theoretical treatments.
> 2. we have removed the analysis to which you are referring.
> 3. we have attempted to be more clear about what we mean by “graded symbols” in section  at the end of the first paragraph in the first paragraph of the Discussion. We have expanded upon what we mean by symbolic gradience in the second paragraph of Results section 4.3. Specifically, we have added,
>
> "We see a gradience in the IIA, where the interventions have a relatively smooth decrease in IIA when the quantities involved in the intervention are large and when the intervention quantities have a greater absolute difference. This indicates that the neural variables possess some level of graded continuity. We refer to such neural variables as symbol-like, or graded neural variables."
>
>
> Thank you again for your feedback, it has made the paper much stronger.

---

> > ### Comment · Reviewer_GPsF · 2025-04-21
> >
> > Thank you for incorporating my suggestion to restructure the results section. I’ve revised my recommendation.

---

### Review · Reviewer_co4F · 2025-03-16

**Summary Of Contributions:**

This paper investigates whether neural networks encode symbol-like numeric representations when trained on numeric equivalence tasks using next token prediction. Through causal interventions (DAS), the paper shows that recurrent neural networks use a cumulative counting strategy, while transformers recompute numeric values at each step. The paper also discusses graded neural representations and architectural differences, suggesting that neural networks approximate but do not fully internalize symbolic computation.

**Audience:**

Yes

**Claims And Evidence:**

Yes

**Requested Changes:**

1.	Modify the description of the numerical equivalence task to make it clearer.

2.	Provide more justification or ablation study on how certain hyperparameters are chosen.

3.	Explain why recurrent neural networks have a low alignment accuracy on *Same-object* tasks.

4.	Please give any comments (or revisions, if possible) to Weaknesses 4-6.

5.	The overall layout of the figures can be improved. Figure 2 and Figure 3 appear early in the main text, but are not referenced until the last two pages.

6.	I’m still confused about what Figure 3 (a) and (b) try to express. It is better that the authors go through them step by step, with detailed explanations on the legends.

**Strengths And Weaknesses:**

Strenghts:

1.	The paper focuses on an important problem: whether and how symbol-like representations emerge in neural networks when trained on tasks that require symbolic reasoning.

2.	Most parts of the paper are well structured and clearly written.

3.	The paper provides detailed experimental settings, which enhances reproducibility.

Weaknesses:

1.	In Section 3.1, the description of the numerical equivalence task can be made clearer. A step-by-step example may help. For instance, “We consider the Single-object task and choose the object quantity to be 3. Then, the input sequence to the model is [BOS, D, D, D, T]. Given this input, the model is expected to output the same number of response tokens R as the number of demo tokens D presented in the input. Therefore, the expected completion is [R, R, R, EOS].” Furthermore, I’m still not sure whether the demo tokens remain the same across different input sequences. For example, in Single-object tasks, will there be two input sequences [BOS, D, D, D, T] and [BOS, E, E, E, T], where D and E are both considered demo tokens?

2.	The justification for choosing certain hyperparameters (e.g., the number of replaced dimensions in DAS) could be stronger. Currently there is no ablation study on how different numbers of replaced dimensions in DAS would affect the result.

3.	In Figure 4, the paper finds that recurrent neural networks usually have a low alignment accuracy on *Same-object* tasks, but the reason is not well explored.

4.	The DAS method requires one to manually curate a set of symbolic algorithms. Only with those manually curated algorithms at hand, one can then measure the alignment between the neural representation and the symbolic algorithm. This approach is potentially difficult to scale up, as the symbolic algorithms are crafted by humans and usually takes some effort. It would be better if this method could be extended to directly identify whether the neural network implements a specific symbolic algorithm.

5.	The DAS method only roughly measures the alignment between the neural representation and a specific symbolic algorithm, but it does not identify the circuit that implements the algorithm. In other words, taking transformers as an example, the method does not tell us which model component (An attention head? Or an MLP?) is responsible for computing the addition and subtraction operation, and which part of the hidden states is responsible for storing intermediate results. This limits the significance of the discovery presented in this paper.

6.	The study does not explicitly quantify how much of the network’s performance is explained by these symbolic representations. It is good that the authors analyze the correlation between the alignment to the symbolic algorithm and network’s performance, but it should be noted that this analysis is correlational rather than causal.

7.  The task and the models are relatively simple. The paper might be more impactful by considering more complex numerical tasks.

---

> ### Author Response · Authors · 2025-04-14
>
> Thank you for your thoughtful review, it was very helpful for improving the paper. We address the issues you raised as follows:
>
> 1. We added a complete description of the possible token sets for each task in section 3.1 and we added expanded examples.
> 2. We added Figures 8 and 9 to the supplement to show the effect of changing the size of the intervened subspace in DAS trainings.
> 3. We include the Same-Object task mainly as a means to demonstrate the non-triviality of our results using DAS. We have introduced a generalization of DAS, however, to allow for more flexible alignment functions. Using these relaxed alignment functions, we have trained linear alignments that define a linear relationship between the neural activity and the interpretable variables from a symbolic algorithm. We argue that this serves as a form of understanding the model’s neural activity, by relating/aligning the activity to interpretable variables.
> 4. As mentioned in point 3, we have introduced the notion of Alignment Functions to address some of the limitations of DAS.
>
> We appreciate your feedback on point 5. We have included a theoretical analysis of a 1-layer NoPE transformer in section 4.2.2 for greater clarity on the transformer solutions. Additionally, we note that DAS is a general framework for performing causal analyses on neural subspaces. It can be applied to any representation in the NNs. We have added a new Figure 3 to show how the DAS alignment can be used to identify and interpret neural activity in the original neural space. We note, however, that there may be stronger explanatory power in treating the neural variable’s subspace as the fundamental unit (as opposed to individual neurons). We say this in consideration of population encoding. We show in Results section 4.1.5 that the Linear Alignment functions allow DAS to find and use these subspaces even when they are non-orthogonal.
>
> We are not sure that we understand point 6. Are you asking whether the symbolic alignments are necessary (and not just sufficient) for network performance? This is an interesting question that we are exploring in ongoing work. We are also unsure what you mean by the claim that “this analysis is correlational rather than causal.” We causally test the model’s alignment to the symbolic algorithms through the interchange interventions and measure the alignment through the models’ corresponding interchange intervention accuracy (IIA). The activations are causally changed in such a way so as to result in meaningful counterfactual behavior. If the interchange interventions had no causal effect, the models’ near-perfect original accuracy demonstrates that the models would simply predict the original sequence instead of the counterfactual sequence. If the interventions had a random effect, the models would not predict the counterfactual sequences with such high accuracy. We can make this claim because random network performance on this task and early DAS training steps perform near 0% (due to the necessity of both the correct number of R tokens followed by the EOS token in the response phase).
>
> 5. We have reorganized and changed many of the figures.
> 6. We have replaced Figure 3 with a new figure.
>
> Thank you again for your helpful review!

---

> > ### Comment · Reviewer_co4F · 2025-05-11
> >
> > I'd like thank the authors for the response and revision. And sorry for my late reply. I will consider the revision when making the recommendation.
> > - The revised Section 3.1 now seems much clearer than the previous one.
> > - The increase of IIA with subspace dimension seems reasonable.
> > - I appreciate the effort to propose a more generalized and relaxed family of "alignment functions," but it does not address my fundamental concern for the need for manual curation of a set of symbolic algorithms in point 4. Human effort is always required to obtain the ground-truth symbolic algorithms before we can move on with the DAS algorithm.
> > - I appreciate the analysis on a simple 1-layer NoPE transformer, thanks for adding that.
> > - To clarify, point 6 refers to the analysis in Section 4.3 and Figure 5(a) in the original manuscript. By "causal" I mean the relationship between alignment score (IIA) and model accuracy, not the way you perform intervention and compute IIA. To be specific: if we intervene to increase IIA (using whatever method), will the model accuracy also increase accordingly? And will the model accuracy decrease if we decrease IIA? This shows a causal relationship between IIA and model accuracy beyond simple correlation.
> > - Figure 3 is still confusing.

---

> > > ### Author Response · Authors · 2025-05-14
> > >
> > > Thank you for your response, we are at fault for the initial delay!
> > >
> > > - In response to concerns about the manual curation of symbolic algorithms, yes, that is still a weakness of the DAS method. We leave automated approaches to future work.
> > > - In response to the point about the relationship between the model's accuracy and IIA, all of our analyses are performed on models with frozen weights. So, the model's accuracy does not change when we improve IIA. Perhaps you are pointing out that the IIA could be higher than the model's original accuracy when using models with imperfect accuracy? In this case, our results showing the IIA over the course of the models' training somewhat address this point in that they show IIA and accuracy correlate even when the model has low task performance.

---

### Author Response · Authors · 2025-03-28
**Thanks for the useful feedback - restructuring and responses to specific reviewer comments underway.**

We appreciate the comments of the reviewers and we are in the process of restructuring the manuscript as well as performing several specific further investigations and clarifying our arguments, as requested by reviewers co4F and gPsF.  The main restructuring will be to present the analyses of the RNN and transformer models in separate sections since the experiments and interventions are largely distinct between the two cases.  We believe these edits will help bring out what we see as one of the important contributions of our work, which is that it sheds light on the quite distinct solutions to our task that are learned by the RNNs studied on the one hand and the transformers on the other.

We have been granted a two week extension to complete the revision of the ms and we will be writing separate responses to each reviewer as well as a brief overall characterization of the revised ms when we upload the revision for re-review by the new deadline of about April 13.

Thanks again for the helpful feedback, which will strengthen the final version of our paper.

---

### Author Response · Authors · 2025-04-14
**Extensive Changes**

We thank the reviewers for their feedback and we apologize for the delay in our response. We have made substantial changes to the paper. In addition to reorganization, we have added a number of theoretical and empirical analyses, we have improved on the existing analyses, and we have provided a new variant of DAS.

We hope that these changes have made the paper clearer, better, and more impactful. We look forward to the reviewer’s feedback.

---

### Author Response · Authors · 2025-07-11
**Updated draft to address minor revisions, part 1**

Thank you to the reviewers and action editor for substantially improving the quality of our initial submission. We hope that the following revisions will enable the action editor to reccommend acceptance.

**1.** To address request 1, we first include an excerpt in the abstract:

> We extend our analytic toolkit to address the failure cases by expanding the DAS framework to a broader class of alignment functions that more flexibly capture NN activity in terms of interpretable variables from SAs—we provide theoretic and empirical explorations of linear alignment functions in contrast to the preexisting orthogonal alignment functions.

We have also edited contribution 2 in the introduction:

> We show that LAFs can be used to control NN behavior through their neural activity aligned to interpretable SAs, and we provide a theoretical analysis of LAFs and Orthogonal Alignment Functions (OAFs) to better understand their properties.

We then expand upon the LAF results in section 4.1.1 where we include reference to supplemental PCA figures 16 and 17. We state that,

> In accordance with the OAF IIA and what we observed in Figure 17, we do not conclude that the model has distinct Count and Phase variables in the same way as the Multi-Object models. It does mean, however, that such variables are recoverable from its state space via an invertible linear mapping, and can be causally intervened upon after such recovery, before being mapped back into the state space of the model to produce desired, predictable behavior.

We then address the fact that the LAF has high IIA for both the Up-Down and Up-Up algorithms with the following in section 4.1.1,

> We also see in Figure 2 that the GRU LAF alignments reach very high IIA (>95%) for the Demo Count and Resp Count variables of the Up-Up program. It appears that the flexibility of the LAFs allows the states of networks to be aligned with conceptually distinct programs. In this case, it is possible to compose the Count variable of the Up-Down program using the difference between the Demo Count and Resp Count variables from the Up-Up program. This perhaps explains why both LAF alignments result in high IIA for both the Multi- and Same-Object tasks as shown on the right side of Figure 2. We provide an additional SA with corresponding OAF and LAF alignments in Supplemental Section A.3 as a baseline demonstrating that LAFs do not have high IIA for all possible SAs.

We summarize our claims about LAFs in the final paragraph of section 4.1.1,

> LAFs can allow for causal interventions of high accuracy under a wider range of circumstances than OAFs, though, this measure alone becomes less diagnostic of the specific SA that best aligns with the algorithm that the neural network has learned.

And we refine our intuitions and claims in section 4.1.5,

> This means that the LAF can allow the intervened subspaces to be non-orthogonal in the original neural space, thus allowing the LAF to formulate each variable from the SA in linear terms of the other variables.

**2.** We have moved what was previously figure 3 to now be figure 4 so as to be closer to an additional section 4.1.6 which specifically focuses on the figure.

We also remove direct reference to superposition and instead focus on the general notion of distributed coding by removing the excerpt, "We leave to future work explorations on how alignment functions can be used to understand NN solutions that use informational superposition (Elhage et al., 2022; Olah, 2023)." from section 4.1.5, and add to the end of section 4.1.6,

> This demonstrates a way to view the causally relevant neural activity in the original neural space, through the lens of the interpretable variables. We can see that many of the raw neurons play a causal role in both the Phase and Count neural coding, confirming notions of distributed coding from prior work (Smolensky, 1988; Elhage et al., 2022; Olah, 2023)."

**3.** We now explicitly define "graded neural variables" in contribution 1 as follows:

> neural variables are defined as representational subspaces that causally align with a variable from an SA, and they are "graded" or "symbol-like" when they exhibit signatures of a continuum rather than being fully discrete.

and in section 4.1.3:

> This indicates that the neural variables have some level of graded continuity despite the task using fully discrete numeric values.  We refer to such neural variables as \emph{graded} or \emph{symbol-like}. This notion of graded, symbol-like variables can be contrasted against cases where the errors are uniformly distributed, independently of the values used in the interventions.

---

> ### Author Response · Authors · 2025-07-11
> **Updated draft to address minor revisions, part 2**
>
> **4.** We have revised claim 4 to now say "confirm" instead of "demonstrate". We defend this with the individual neuron interventions in Section 4.1.2 and Figure 4b (previously Figure 3b).
>
> **5.** We add to section 2 to address background literature:
>
> - To address mapping models, we add the following segment to section 2, "n section 3.4, we introduce generalized alignment functions which can be framed as a specific class of mapping models from Ivanova et al. (2022) where the mapping model in our case defines an invertible, causal relationship between neural subspaces and variables from an interpretable SA, the success of which is measured through NN behavior."
>
> - To address the work of Williams, 2021 on orthogonal vs linear alignments, we have added the following to section 2, "Other works such as Williams et al. (2021) have explored differences between orthogonal and linear relationships between distributed representations where they largely confined their analyses to formally defining metrics on representational dissimilarity."
>
> - To address the requested mech interp works, we have added the following to the beginning of section 2,
>
> >Many prior works have attempted to describe ANNs through SAs. Some relevant examples consist of Lindner et al. (2023) who built a system to compile transformer models from human written code and Michaud et al.  (2024) who perform linear and symbolic regression on simplified NN representations to generate programs that perform like the NNs. Both these cases focus on generating sufficient SAs for network behavior rather than relating the internal mechanisms of the intial NNs to the resulting SAs. Another example is the work of Nanda et al. (2023) who show that small transformers trained on modular addition use discrete Fourier transforms combined with trig identities to solve the task. Nanda et al. use a combination of examinations and ablations to show that the grokked NNs use their proposed solution."
>
> - We have added Heimersheim & Nanda, 2024 to our references when introducing "activation patching"
>
> **6.** We have made a number of minor edits to justify our hyperparameter choices:
>
> - we have added the line "where 20 was chosen to match the human experiments of Pitt et al. (2022)." in section 3.1 to justify sequence lengths
>
> - to justify our held-out object quantities, we have included the line in section 3.1 "We chose 4, 9, 14, and 17 to semi-uniformly cover the space of possible numbers while including even, odd, and prime numbers without affecting model trainings."
>
> - we justify our accuracy threshold in section A.4.1 with the excerpt:
>
> >"We removed models with performance below 99\% to limit our DAS results to perfectly performing models thus simplifying our interpretations of the results. We chose 99\% accuracy instead of 100\% due to slight numerical underflow in accuracy calculations and due the fact that half of the Vary-Length Same-Object models would have been dropped due to low performance."
>
> - we add an expanded exploration of IIA as a function of subspace sizes in Supplemental Figure 9
>
> - We have re-positioned what was previously figure 4 to now figure 3 under section 4.1.3 "Graded Symbols" and previously figure 3 now figure 4 closer to section 4.1.5
>
> **Final Remarks:**
> We have included a few additional changes to the abstract and concluding remarks without changing the core claims of the paper. We hope the reviewers find our finalized changes satisfactory!

---

### Decision · Action_Editor_tsYr · 2025-06-18

**Recommendation:** Accept with minor revision

**Additional Comments:**

Reviewers were in agreement on accepting the work, with the exception of one who took issue with (i) the lack of provided generalizability of the findings, due to the need to hand-craft symbolic algorithms for reference, and (ii) the lack of elaborated discussion of the relationship between accuracy and intervention effect (interchange intervention accuracy; IIA). Since (i) generalizability will not be claimed once Claim (4) is adjusted (as requested below) and (ii) the relationship between accuracy and IIA does not feature in nor affectt the claims, these concerns do not pose a problem for TMLR.

After reviewing the submission in further detail, the Action Editor (AE) requests the following minor revisions:

1. The authors should more clearly state the outcome of investigations performed with the "linear alignment" technique and visualized in Figure 2 (right), and adjust Claim (2b) accordingly. Is the investigation inconclusive? If so, what does mean for LAS? If not, what does it demonstrate?

1. Given that superposition as evidenced in Figure 3 has been demonstrated in prior neural network models (as cited in the submission: Elhage et al., 2022; Olah, 2023), the authors should explicitly state whether their findings here are confirmatory or novel.

1. The authors should clearly define "graded neural variable," which may be defined consequentially as the outcome of a DAS procedure, but needs distinction from (or to be identified with) the classical definition of distributed representation.

1. The authors should adjust Claim (4) about broader importance to neural network interpretability, which cannot be made in the abstract and is not directly supported by the evidence provided. This should be revised to describe the case study in the context of the broader alignment search literature.

1. The authors should include discussion of prior relevant work on mapping models in neuroscience for relating neural activity to ground-truth models ([Ivanova et al., 2021](https://arxiv.org/abs/2208.10668)), orthogonal alignment techniques ([Williams et al., 2021](https://arxiv.org/abs/2110.14739)) and their relaxation to linear alignment, mechanistic interpretation of neural network sequence models with respect to symbolic programs ([Lindner et al., 2023](https://arxiv.org/abs/2301.05062); [Michaud et al., 2024](https://arxiv.org/abs/2402.05110); [Nanda et al., 2023](https://arxiv.org/abs/2301.05217)), and prior work on causal interventions in sequence models via "activation patching" ([Heimersheim & Nanda, 2024](https://arxiv.org/abs/2404.15255)).

1. The authors should provide justification for hyperparameter choices (accuracy threshold, sequence lengths, hold-out selection of 4, 9, 14, and 17), position figures nearer to their mention in the text, and address remaining formatting issues and typos.

Subject to these revisions, the AE can recommend acceptance to TMLR.

**Audience:**

Yes

**Audience Explanation:**

The topic of symbolic interpretation of neural network implementations has a large audience, and this thorough application of several interventional techniques to a new case study will find an audience therein.

**Claims And Evidence:**

Yes

**Claims Explanation:**

This submission uses several techniques including direct alignment search (DAS; [Geiger et al., 2021](https://arxiv.org/abs/2106.02997)) to probe neural network implementations of a counting task that can be realized as high-level symbolic algorithms (symbolic algorithms; SA). The work considers four symbolic algorithms defined over symbolic variables, and investigates whether sequence neural networks code for these symbolic variables in their internal activities.

The papers claims are stated explicitly in the introduction, and can be rephrased as follows:

1. **(a)** Variables in a symbolic algorithm can be used to intervene on activities in sequence neural networks, **(b)** which is evidence that the neural networks code for these variables in their representations, and compute with them.

2. **(a)** The relationship in (1b) is not robust to perturbations of the task, namely whether or not the demo and response tokens are the same type.
**(b)** This can be addressed by generalizing the technique in (1a).

3. The technique in (1a) alongside some theoretical constructions give evidence that certain sequence neural networks use a non-cumulative "anti-Markovian" solution to the counting task.

4. The techniques in (1a) and (2b), alongside other techniques in the paper, are "important" to the field of neural network interpretability.

Claims (1-3) are adequately supported by the DAS alignment results, causal interventions, and theoretical constructions, with one exception: the authors must adjust (2b) to provide explicit interpretation of the inconclusive generalized results (Figure 2, right: no symbolic algorithm differs in its match). Claim (4) regarding broader importance is not supported by the evidence and should be adjusted to describe this case study in the context of the broader alignment search literature.